# Periplaneta americana extract ameliorates recurrent oral ulcers in rats by enhancing the intestinal epithelial barrier and regulating gut microbiota

Kailing Li[1], Liping Yuan[1], Jingyu Zhang[2], Weijun Li[2], Guanhua Zhao[2], Zhongze Chen[1], Yongshou Yang[2], Zhengchun He[2]*, Peiyun Xiao[1]*

1 Engineering Research Center for Development of the Periplaneta americana Industry of Yunnan Provincial Department of Education, Dali University, Dali, China, 2 College of Pharmacy, Dali University, Dali, Yunnan, China

* hezhengchun2006@163.com (ZH); xiaopeiyun@dali.edu.cn (PX)

## Abstract

### Background

Recurrent oral ulcers (ROUs) of oral mucosa disease are difficult to cure and relapse easily. *Periplaneta americana* extract (PAD), a raw material used in Kangfuxin Liquid and Yunnan Baiyao toothpaste, contains a variety of growth factors such as polypeptides and sticky sugar amino acids that promote tissue repair; this can encourage the growth of granulation tissue and reduce inflammation on wound surfaces.

### Objective

In this study, we used a rat model of ROU induced by an antigen emulsifier to assess the ameliorative effects of PAD on rat ROUs and to explore its mechanism of action.

### Methods

The effect of PAD in rats was evaluated by an enzyme-linked immunosorbent assay (ELISA) kit, hematoxylin-eosin staining, immunohistochemistry, reverse transcription quantitative PCR (RT-qPCR), and Western Blot (WB). 16S rRNA sequencing and gas chromatography-mass spectrometry (GC-MS) were used to detect the changes in intestinal flora and its metabolite short-chain fatty acids (SCFAs) in the feces of rats, respectively.

### Results

PAD significantly reduced the infiltration of local inflammatory cells and significantly downregulated interleukin (IL)-6 and tumor necrosis factor (TNF-α), while upregulating IL-2, IL-10, and vascular endothelial growth factor (VEGF). In addition, PAD

**Data availability statement:** All relevant data are within the manuscript and its Supporting information files.

**Funding:** This study was supported by the Basic Research Key Projects of Science and Technology Department of Yunnan Provincial (202501AS070162) , the Joint Special Focus Program for Local Colleges and Universities in Yunnan Province (grant number 202401BA070001-007), and the Yunnan Expert Workstation (202405 AF140044).

**Competing interests:** The authors have declared that no competing interests exist.

altered the diversity and abundance of the gut microbiota and increased fecal short-chain fatty acid levels. Notably, PAD can improve the pathological injury of the colon, enhance the intestinal barrier of the colon, and reduce the apoptosis of colon cells.

## Conclusion

The oral administration of PAD may effectively treat ROUs via regulating metabolites and altering the composition of the intestinal microbiota, thereby improving the intestinal barrier function.

---

## 1. Introduction

Recurrent oral ulcers (ROUs) are also called recurrent aphthous ulcers and recurrent aphthous stomatitis [1,2]. According to an epidemiological survey, the incidence of ROUs ranges from 5% to 60% [3]. While ROU attacks are localized and typically resolve on their own, the intense burning pain, swollen lymph nodes, general malaise, and other symptoms that accompany these attacks still interfere to some extent with the patient's daily life and have an impact on their quality of life [4]. Currently, ROU is clinically treated with Western medicine, and the commonly used drugs are glucocorticoids, thalidomide, and colchicine, but they have adverse effects such as teratogenicity, dizziness, and gastrointestinal adverse reactions [5]. Traditional Chinese medicine offers the unique advantage in ROU treatment by not only effectively promoting ulcer healing and relieving pain, but also significantly inhibiting ROU recurrence rate [4,6–8]. Thus, increasing number of patients are seeking traditional Chinese medicine for ROU treatment.

*Periplaneta americana* (*P. americana*), commonly known as cockroaches, is member of the class Insecta, order Blattaria, and family Blattidae. Extensive pharmacological studies on *P. americana* extracts have revealed that it promotes tissue repair [9–11], immunoregulation [12], and exerts anti-oxidation [13,14] effects. The combination of Kangfuxin liquid and oral ulcer powder may help reduce inflammatory factors in patients with recurrent mouth ulcers, resulting in less pain and a lower recurrence rate of recurrent mouth ulcers [15]. A retrospective analysis showed that Kangfuxin liquid reduced inflammatory responses, decreased oral mucosal reactions and oral pain, and promoted cellular immune function, which in turn reduced the incidence of radiotherapy-induced oral mucositis in patients with squamous cell carcinoma of the head and neck [16]. In addition, Kangfuxin liquid can also effectively prevent oral mucositis caused by radiotherapy in nasopharyngeal cancer patients, reduce the severity of oral pain, and then improve the quality of life of patients [17]. Our previous study on *P. americana* extract (PAD), a raw material of the Yunnan Baiyao 'active peptide toothpaste' (industry standard number: QB/T5287-218.), showed that it accelerated the healing of oral ulcer surfaces in rats. However, the repair mechanism of PAD on ROU remains unclear and warrants further study.

The oral cavity is the starting point of the digestive tract and connects to the gastrointestinal tract. Some pathogens existing in the mucosa of the oral cavity

and digestive tract can be transferred between the locations in the gastrointestinal tract; therefore, there is a correlation between the occurrence of ROUs and the digestive tract [18,19]. A higher prevalence of gastrointestinal diseases has been noted in individuals with ROUs than in those without, and the episode frequency and symptom severity of ROUs also tend to increase with an increase in the prevalence of gastrointestinal diseases, thereby indicating a close relationship between the two [20,21]. Furthermore, the occurrence of ROUs may be associated with the imbalance of gut microbiota (GM) and its metabolites (short-chain fatty acids, SCFAs) in feces, and an imbalanced GM can lead to the destruction of the intestinal mucosal barrier (IMB) and immunological derangement, thereby promoting the release of inflammatory factors such as interleukin (IL)-2, IL-6, IL-10, and tumor necrosis factor (TNF)-α and inducing the onset of chronic inflammation; collectively, these processes can aggravate ROU symptoms [22–25]. Therefore, studying the relationship between ROUs and intestinal microecology can provide theoretical basis for ROU treatment. To the best of our knowledge, this is the first study to explore and report the therapeutic mechanism of PAD on ROU based on intestinal microecology.

## 2. Materials and methods

### 2.1. Materials and reagents

*P. americana* (batch number: 20210311) was purchased from Yunnan Jingxin Biotechnology Co., Ltd. (Yunnan, China). Prof. Zizhong Yang from Yunnan Key Laboratory of Insect Biomedicine Research and Development of Dali University identified it as *P. americana*. Levamisole hydrochloride (LM) tablets (batch number: 200201) were obtained from Shandong Renhetang Pharmaceutical Co., Ltd. (Shandong, China). Complete Freund's adjuvant, acetic acid, propionic acid, butyric acid, and pentanoic acid were purchased from Sigma-Aldrich (St. Louis, MO, USA). Vascular endothelial growth factor (VEGF), IL-2, IL-6, IL-10, TNF-α, and secretory immunoglobulin A (SIgA) ELISA kits were purchased from Nanjing Jiancheng Biology Co. Ltd (Nanjing, China). A DNA Extraction Kit was purchased from Omega Bio-Tek (Doraville, GA, USA). An AxyPrep™ DNA Gel Recovery Kit was purchased from Axygen (Hangzhou, China). Primer sequences were synthesized by Shanghai Bioengineering Co., Ltd. (Shanghai, China). A BCA Protein Assay Kit was purchased from Beijing Solarbio Science & Technology Co., Ltd. (Beijing, China). The polyvinylidene fluoride membrane was purchased from Millipore (Billerica, MA, USA). 5% bovine serum albumin blocking solution was purchased from Biofroxx (Frankfurt, Germany). GAPDH and bax were purchased from Cell Signaling Technology (Danvers, MA, USA). Occludin and zonula occludens-1 (ZO-1) were purchased from Invitrogen (Carlsbad, CA, USA). Claudin-1 and horseradish peroxidase-labeled goat anti-rabbit secondary IgG antibody were purchased from Proteintech Group Inc. (Wuhan, China). Bcl-2 and caspase-9 were purchased from Abcam (Cambridge, UK). The developer solution was purchased from Dalian Meilun Biotechnology Co. Ltd. (Liaoning, China).

### 2.2. Extraction of PAD

PAD was prepared using an extraction method previously standardized by our research group. Briefly, 540 g of *P. americana* (medicinal materials) was crushed into a powder using a herbal pulverizer and passed through a size 24 mesh and then extracted 3 times using an alcohol-water (60:40) mixture for 3 h each time at 70–80 °C. Next, the extraction solutions were filtered, combined, concentrated under reduced pressure, and then degreased. The defatted solution was adsorbed using activated carbon column chromatography and eluted with an alcohol-water (80:20) mixed solvent. Finally, the eluate was collected, concentrated under reduced pressure, and freeze-dried to obtain about 32g of PAD with an extraction yield of about 6%.

### 2.3. Experimental animals

Seventy-eight male specific-pathogen-free Sprague-Dawley rats, weighing approximately 200 g, were purchased from Hunan Silaikejingda Experimental Animal Co. Ltd. (Hunan, China). The experimental animal license number was SCXK

(Xiang)-2019−0004. The Dali University Laboratory Animal Ethics Committee gave its approval to the experimental proto-col (approval number: 2019-PZ-078). The rats were kept at 21±2 °C and a relative humidity of 45%–65%, with a 12/12 h light/dark cycle. They were allowed to freely access food and water.

## 2.4. Establishment of an ROU model and grouping of the rat

After 7 d of adaptive feeding, the rats were fed a standard laboratory diet and water *ad libitum*. The oral mucosal tissue homogenates were prepared using 30 rats, and 48 rats were randomly assigned to six groups based on body weight ($n = 8$ per group): control, model, Levamisole hydrochloride (LM), PAD-Low dose (PAD-L), PAD-Medium dose (PAD-M), and PAD-High dose (PAD-H). The 30 rats were humanely euthanized by intraperitoneal injection of sodium phenobarbital [26], and an appropriate amount of saline was injected into the oral mucosa of the rats with a syringe to separate the mucosal tissue from the connective tissue. The oral mucosal tissue was stripped from the rats, the residual connective tissue was removed and rinsed with saline, and filter paper absorbed the water. The dissected oral mucosal tissue was immediately placed in a −80 °C refrigerator. Oral mucosal tissues frozen in the −80°C refrigerator were taken out and cut with scissors, added with PBS (pH = 7.4), and then ground at 4°C with a tissue grinder to make an oral mucosal tissue homogenate. An equal volume of oral mucosal tissue homogenate was mixed with complete Freund's adjuvant, and the antigen emul-sifier was prepared by mixing the two thoroughly with a vortex oscillator to form an emulsion until no delamination was observed. The rats (except the control group, which were injected with complete Freund's adjuvant) were hypodermically injected with 0.1 mL antigen emulsifier on both sides of the spine, once a week, for 6 weeks, to establish a rat model of immune ROUs. After successful modeling, the control and model groups were administered normal saline, the LM group was administered LM tablets (20 mg/kg), and the PAD-L, PAD-M, and PAD-H groups were treated with 0.3125, 0.6250, and 1.250 g/kg PAD, respectively, by gavage once a day, for 7 d. In this study, the high dose was mainly based on the dose of Kangfuxin liquid for human clinical use, which was converted into the equivalent dose of rats according to the body surface area. The coefficient was 6.25 for a person weight of 60 kg.

## 2.5. Preparation of biological samples

Fresh fecal samples were collected from each rat on the day before the end of treatment. One part of fecal samples was frozen at –80 °C for subsequent gas chromatography-mass spectroscopy (GC-MS) analysis, while the other was used for 16S rRNA determination.

After 24 h of fasting, the rats were humanely euthanized by intraperitoneal injection of sodium phenobarbital. Through-out the research, all animals remained in good health, and no unexpected deaths occurred. The rats were subsequently disposed of in accordance with the guidelines set forth by the National Research Center's Safety and Health Committee (NRC) [26]. Blood samples were collected from the abdominal aorta of the rats and incubated on ice for 1 h. After centrifu-gation (2000 rpm, 10 min), the serum obtained was used for biochemical index detection.

Oral mucosa and colon tissues were removed immediately after the rats were sacrificed. The volume of the oral mucosa tissue was about 0.5×0.5×0.2 cm, while the colon tissue length was 2–3 cm. To get prepared for histopathology, the tissues were washed with normal saline, dried with filter paper, and preserved in 4% paraformaldehyde solution for 24–48 h. The remaining oral mucosa and colon tissues were stored at –80 °C for subsequent detection of biochemical indices.

## 2.6. Histological analysis

Rat oral mucosa and colon tissue samples were dehydrated, paraffin-embedded, cut into slices 5 μm thick, and then baked at 60 °C for 1 h. The tissue sections were viewed and photographed under a microscope (Olympus, Tokyo, Japan) after being stained with hematoxylin and eosin (H&E).

## 2.7. Enzyme-linked immunosorbent assay (ELISA)

Serum levels of VEGF, IL-2, IL-6, IL-10, and TNF-α and SIgA in the colon tissue were measured using ELISA kits according to the manufacturer's instructions.

## 2.8. Measurement of short-chain fatty acid (SCFA) levels in the intestinal feces

Accurately 0.2 g fecal samples stored at –80 °C was finely blended with 1 mL methanol to prepare fecal suspension. Next, an appropriate amount of concentrated sulfuric acid was added to the suspension, to adjust the pH value to 2.0–3.0. This mixture was then centrifuged (5000 rpm, 20 min) at 25 °C to obtain the supernatant. A filter membrane with a 0.45-μm microporous surface was used to filtrate the supernatant, and the filtrate was then collected for GC-MS (Agilent Technologies Inc., Santa Clara, CA, USA) analysis [27]. Acetic, propionic, butyric, and pentanoic acids were precisely weighed and mixed into a standard solution. The Agilent 7890A GC-MS system with Agilent DB-WAXDB-WAX (30 m × 250 μm, 0.25 μm) column was used to determine SCFA content. Helium was used as the carrier gas, at a flow rate of 0.5 mL/min, and a 1-μL filtrate was infused into the sample, at a split ratio of 10:1. The initial temperature of the column was 100 °C and allowed to stabilize for 2 min; the temperature was then raised to 180 °C at 5 °C/min and maintained for 2 min. The solvent delay was 3 min. The temperature of the Agilent 5975C-MSD detector was set at 230 °C. The temperatures of the inlet, ion source, and transmission line were 230 °C, 250 °C, and 230 °C, respectively. The electron bombardment energy was 70 eV.

## 2.9. Gut microbiota sequencing analysis

The use of a fecal genomic DNA Extraction Kit was employed to extract genomic DNA from the rat fecal samples, and 1% agarose gel electrophoresis was utilized to determine the DNA concentration and purity. The V3–V4 region of the bacterial 16S rRNA was amplified using PCR and specific primers with the barcode as the amplification primer. The amplified products were detected using 2% agarose gel electrophoresis, and the target bands were recovered using an AxyPrep™ DNA Gel Recovery Kit. The PCR products were quantified using a QuantiFluor™-ST blue fluorescence quantitative system (Promega, Madison, WI, USA). The Illumina PE250 library was prepared using RNA products, and high-throughput sequencing was performed using the Illumina MiSeq platform (Shanghai Yuan Shen Biomedical Technology Co. Ltd., Shanghai, China).

The PE reads obtained from the Illumina PE250 sequencing were screened and filtered, and cluster analyses of the operational taxonomic units (OTUs) and species taxonomic analysis were carried out according to a sequence similarity of 97%. Based on the cluster analysis results of the OTUs, the richness and uniformity of the community in the OTUs were analyzed using correlation indices (Chao1, Ace, Shannon, and Simpson indices), and sequencing depth was detected. The community structure was statistically analyzed at various taxonomic levels. Differences in species abundance and diversity among the samples were analyzed using LefSe, in combination with the above analysis.

## 2.10. Reverse transcription quantitative PCR (RT-qPCR) analysis

Total RNA was collected from the oral mucosa and colon tissues of the rats using the TRIzol™ method, and its purity and concentration were determined using an ultramicro nucleic acid and protein analyzer (Thermo Fisher Scientific, Waltham, MA, USA). RNA was reverse-transcribed using a Reverse Transcription Kit (catalog no. 00906776, Thermo Fisher Scientific) and amplified using PCR on a CFX96 Real-Time PCR Detection System (Bio-Rad Laboratories, Hercules, CA, USA). The primer sequences used for PCR amplification were obtained from the NCBI database (http://www.ncbi.nlm.nih.gov/). Primers for IL-6, IL-10, TNF-α, VEGF, occludin-1, ZO-1, claudin-1, bcl-2, bax, and caspase-9 were designed using the Oligo Design software (Table 1). All primer specificities were confirmed using the NCBI BLAST software. GAPDH was used as an internal control [28]. The data were analyzed using the $2^{-\triangle\triangle Ct}$ method [29].

**Table 1. Designed primer sequences.**

| Gene name | | Primer sequence 5′-3′ | Primer length |
|---|---|---|---|
| IL-2 | F | CACTGACGCTTGTCCTCCTT | 193 |
| | R | TGTTTCAATTCTGTGGCCTGC | |
| IL-6 | F | GAGCCCACCAGGAACGAAAGTC | 126 |
| | R | GGGAAGGCAGTGGCTGTCAAC | |
| IL-10 | F | GTCCACTCGCAAGGGCAGAAAG | 95 |
| | R | TCTAACTGGCAGAGGAGGTCACA | |
| TNF-α | F | AAAGGACACCATGAGCACGGAAA | 136 |
| | R | CGCCACGAGCAGGAATGAGAAG | |
| VEGF | F | CTGCTGTGGACTTGAGTTGG | 119 |
| | R | CAAACAGACTTCGGCCTCTC | |
| Occludin | F | GTCTTGGGAGCCTTGACATCTTG | 174 |
| | R | GCATTGGTCGAACGTGCATC | |
| ZO-1 | F | GCCACACTGTGACCCTAAAAC | 95 |
| | R | ACAGTTGGCTCCAACAAGGT | |
| Claudin-1 | F | CTGTCCCCGGAAAACAACCT | 67 |
| | R | CCCACTAGAAGGTGTTGGCT | |
| Bcl-2 | F | GGACTGGGTGAGAAACGAGC | 94 |
| | R | TTTCCGGCTCTTGTGGAAGC | |
| Bax | F | AGCGAGACCTGGAGCAAGCC | 121 |
| | R | GCACTGTCACCTGGAAGCAGAG | |
| Caspase-9 | F | GGACACCATGAGCACGGAAAGC | 176 |
| | R | CGCCACGAGCAGGAATGAGAAG | |
| GAPDH | F | GGGGCTCTCTGCTCCTCCCTG | 107 |
| | R | CGGCCAAATCCGTTCACACCG | |

## 2.11. Immunohistochemistry analysis

Paraffin-embedded colon tissue sections were incubated overnight with primary antibodies against occludin, ZO-1, and claudin-1 (at dilutions of 1:250 or 1:500) at 4 °C. The slices were then incubated with a horseradish peroxidase-labeled goat anti-rabbit IgG secondary antibody, at room temperature for 60 min, developed with 3,3'-diaminobenzidine, stained with hematoxylin, and observed and photographed under a microscope (Olympus, Japan).

## 2.12. Western blot analysis

Radioimmunoprecipitation assay lysate buffer was added to the rat colon tissue, and the total protein concentration was measured using a BCA Protein Assay Kit. Proteins were separated using sodium dodecyl sulfate-polyacrylamide gel elec-trophoresis and transferred to a polyvinylidene fluoride membrane. The membrane was blocked for 2 h using 5% bovine serum albumin blocking solution and incubated with primary antibodies against GAPDH (1:1000), occludin (1:500), ZO-1 (1:500), claudin-1 (1:400), bcl-2 (1:1000), bax (1:1000), and caspase-9 (1:1000), at 4 °C overnight. Thereafter, the mem-branes were incubated with a horseradish peroxidase-labeled goat anti-rabbit secondary IgG antibody, at 25±5 °C for 2 h. A developer solution was used according to the manufacturer's instructions, to detect the immunoreactive bands on the PVDF membrane. GAPDH was used as an internal control.

## 2.13. Statistical analysis

The gray value of the gel bands was quantified using ImageJ software; the gray value of the target protein/GAPDH was used for the relative quantification of the protein. Image-Pro Plus 6 software (Media Cybernetics, Inc., Rockville, MD, USA) was used to quantify the positive expression following immunohistochemistry, and the integrated option density/area was considered as the relative expression level of the protein.

SPSS 22.0 statistical software (SPSS Inc., Chicago, IL, USA) was used for data analysis. The results are expressed as mean ± SEM ($\bar{x} \pm s$), and differences between groups were analyzed using one-way analysis of variance. Results with $p < 0.05$ were considered statistically significant.

## 3. Results

### 3.1. *Periplaneta americana* extract (PAD) improved the number, area, and duration of ulcers in recurrent oral ulcers (ROU) rats

In the model group, white ulcers of different sizes appeared on the buccal mucosa of both sides of the mouth, and the mucosa was congested and red and lasted for a long time. Compared with the model group, the number, diameter, and duration of ulcers in each treatment group were significantly reduced ($p < 0.01$). In addition, ulcer number, diameter, and duration tended to decrease in a dose-dependent manner with increasing PAD concentration. The experimental results are shown in Table 2. It is suggested that PAD can effectively improve the ulcer condition of ROU rats.

### 3.2. Histopathological observation of the oral mucosa in rats

The oral mucosa of rats from different groups was subjected to H&E staining and histopathological analysis. The oral mucosa epithelial tissue of the control group had an intact structure, and the cells of the basal layer and lamina propria were arranged regularly. However, the oral mucosa epithelial tissue of the ROU rats showed partial destruction and thinning, irregular arrangement of cells in the basal layer/lamina propria, and infiltration of multitudinous inflammatory cells. The oral mucosa epithelial tissue of the PAD-M and PAD-H groups was gradually repaired and tended to be complete, with no obvious inflammatory cell infiltration, whereas in the PAD-L and Levamisole hydrochloride (LM) groups, there were no significant defects in the oral mucosa epithelial structure, with improved inflammatory cell infiltration (Fig 1A). These experimental results showed that PAD effectively reduced the antigen emulsifier-induced ROUs in rats.

### 3.3. PAD-mediated regulation of inflammatory cytokines in ROU rats

Serum vascular endothelial growth factor (VEGF), interleukin (IL)-6, and tumor necrosis factor (TNF)-α levels were significantly higher, while IL-2 and IL-10 levels were significantly lower in ROU rats induced with only antigen emulsifier, compared to those in the control rats ($p < 0.05$; Fig 1B). PAD and LM augmented serum levels of VEGF, IL-2, and IL-10, but attenuated serum levels of IL-6 and TNF-α; the therapeutic effect of PAD-H was the most significant ($p < 0.05$).

The expression levels of IL-2, IL-6, IL-10, TNF-α, and VEGF mRNA in the rat oral mucosa were detected using reverse transcription quantitative PCR (RT-qPCR) (Fig 1C), and the results were consistent with serum cytokine expression levels.

**Table 2. Effect of PAD on the number, diameter, and duration of ulcers in a rat model of ROU.**

| Group | Number | Diameter (mm) | Duration (day) |
|---|---|---|---|
| Model | 6.88 ± 1.13 | 3.81 ± 0.70 | 6.38 ± 0.74 |
| LM | 3.88 ± 0.83** | 1.75 ± 0.38** | 3.25 ± 0.89** |
| PAD-L | 5.25 ± 0.71** | 2.94 ± 0.50** | 5.88 ± 0.64 |
| PAD-M | 3.63 ± 0.92** | 1.56 ± 0.42** | 3.13 ± 0.83** |
| PAD-H | 2.25 ± 0.71** | 0.75 ± 0.27** | 2.25 ± 0.71** |

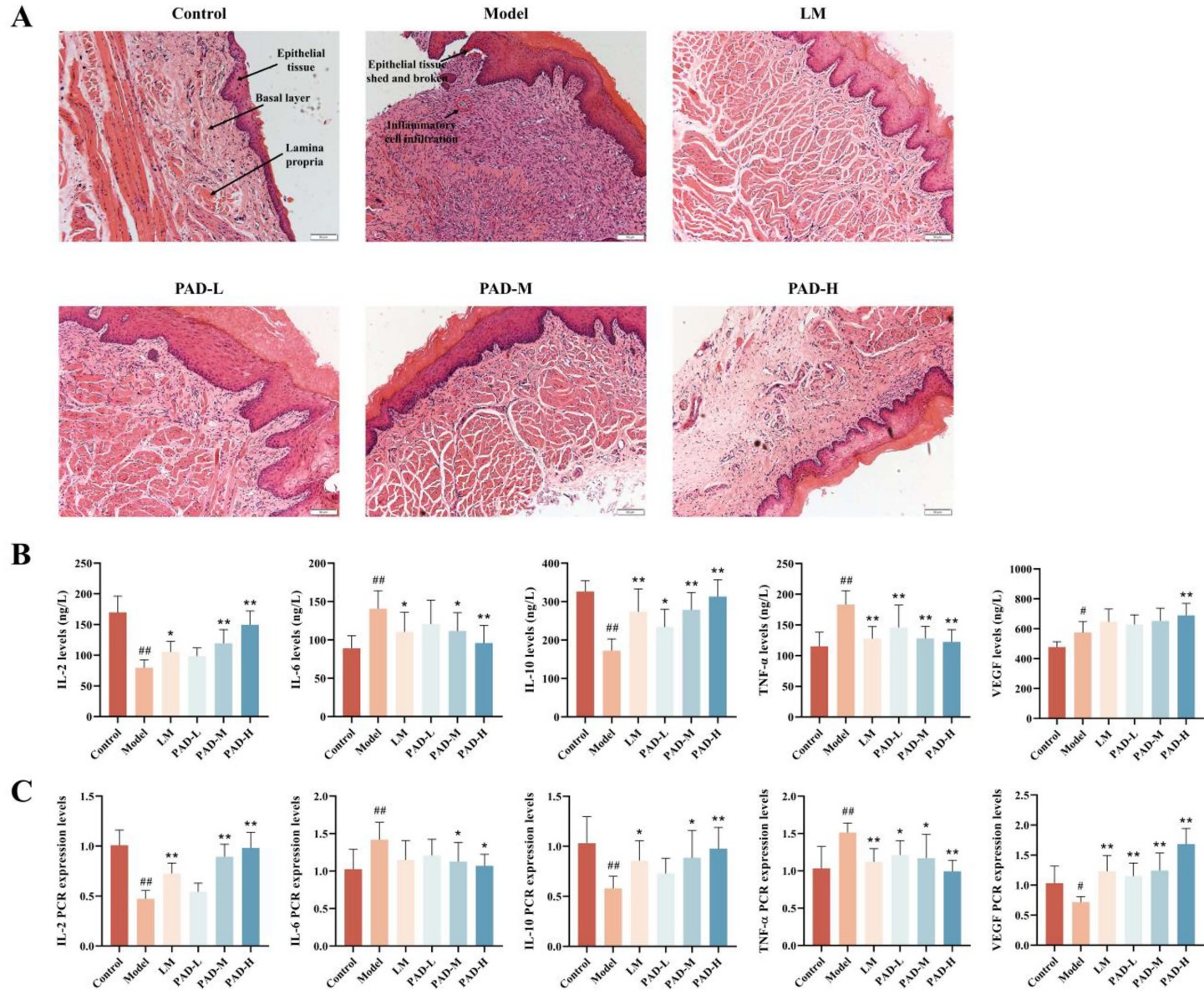

**Fig 1. Effect of PAD on histopathology and expression of inflammatory factors in the oral mucosa of ROU rats. (A)** H&E staining analysis of the oral mucosa of rats (×200). **(B)** Changes in serum levels of IL-2, IL-6, IL-10, TNF-α, and VEGF in rats. **(C)** Effect of the PAD on the expression of IL-6, IL-10, TNF-α, and VEGF mRNA in the rat oral mucosa. Data are expressed as mean±SEM ($n = 8$). **#** $P < 0.05$ and **##** $P < 0.01$, as compared to the control group; **\*** $P < 0.05$ and **\*\*** $P < 0.01$, as compared to the model group.

At the gene level, PAD adjusted the level of inflammatory factors *in vivo* in each group, accelerated the healing of ulcer surfaces, and played a therapeutic role on ROUs.

### 3.4. PAD affects fecal SCFA content in ROU rats

SCFAs are signaling molecules between the main metabolites of intestinal bacteria and the host, and play a key role in regulating the gut microbiota (GM) balance. gas chromatography-mass spectrometry (GC-MS) analysis revealed that

the fecal contents of acetic acid, propionic acid, butyric acid, and pentanoic acid levels were significantly lower in the ROU group, while they were higher in the various PAD dosage groups (Fig 2A, 2B). PAD increased short-chain fatty acid (SCFA) content in rats, thereby indicating that it has a protective effect on the intestinal barrier.

## 3.5. Alpha and beta diversities of microbial 16S rRNA genes

To determine whether PAD could improve the ROUs in rats by regulating the GM, we sequenced the V3–V4 region of the 16S rRNA gene in the feces of rats from each group, to analyze the composition of the GM. We obtained 7431873 high-quality sequences from 48 rat feces samples, with a base number of 3142426527 bp and an average length of 422.83 bp. OTU cluster analysis of species and genera in rat fecal microbial samples showed that the OTU numbers in the control, model, LM, PAD-L, PAD-M, and PAD-H groups were 751, 853, 880, 822, 714, and 711, respectively.

Alpha is a commonly used index to evaluate the microbial community structure. The rarefaction curves (Fig 3A), species accumulation curves (Fig 3B), Shannon–Wiener curves (Fig 3C), and rank-abundance curves (Fig 3D) were close to the saturation state, indicating that the sample coverage was high, and the species had sufficient diversity, reasonable abundance, and good uniformity. The Chao1 and Ace indices reflected the richness of the flora and positively correlated with community abundance. Microbial diversity was mirrored by the Shannon and Simpson indices, in which the Shannon index was positively correlated with community diversity, whereas the Simpson index was not. The Chao1, Ace, and Shannon indices of the GM increased in ROU rats, while the Simpson index diminished. However, PAD restored the Chao1 and Ace indices to approximately those of the control group, whereas there was an increasing trend in the Shannon index and a decreasing trend in the Simpson index (Fig 3E). This indicated that PAD decreased the richness and increased the diversity of the GM in ROU rats.

Beta diversity can be used to reflect the diversity of the microbial communities in different samples. We used Principal Component Analysis to analyze the effect of PAD on the overall structure of the GM in ROU rats. The control and model samples were clustered in two spatial locations, indicating that there were differences in the microbial composition

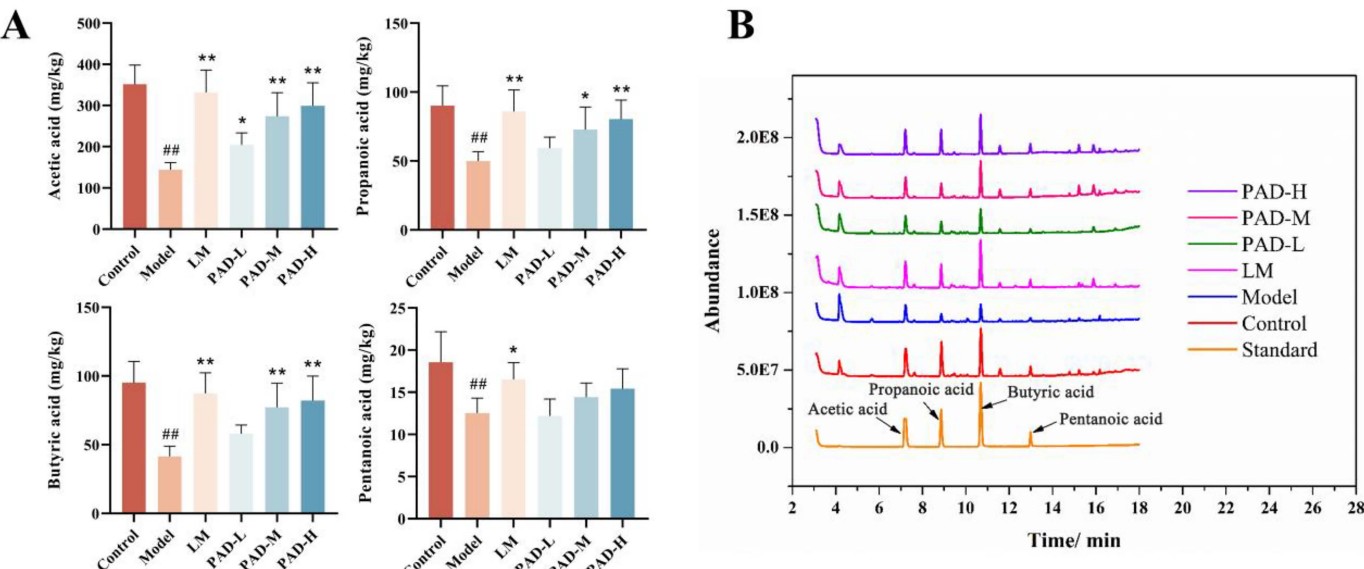

**Fig 2. PAD regulates SCFA content in rat feces. (A)** SCFA content in rat feces. **(B)** Determination of rat fecal SCFA content using GC-MS. Data are expressed as mean ± SEM (*n* = 5). **#** *P* < 0.05 and **##** *P* < 0.01, as compared to the control group; ***** *P* < 0.05 and ****** *P* < 0.01, as compared to the model group.

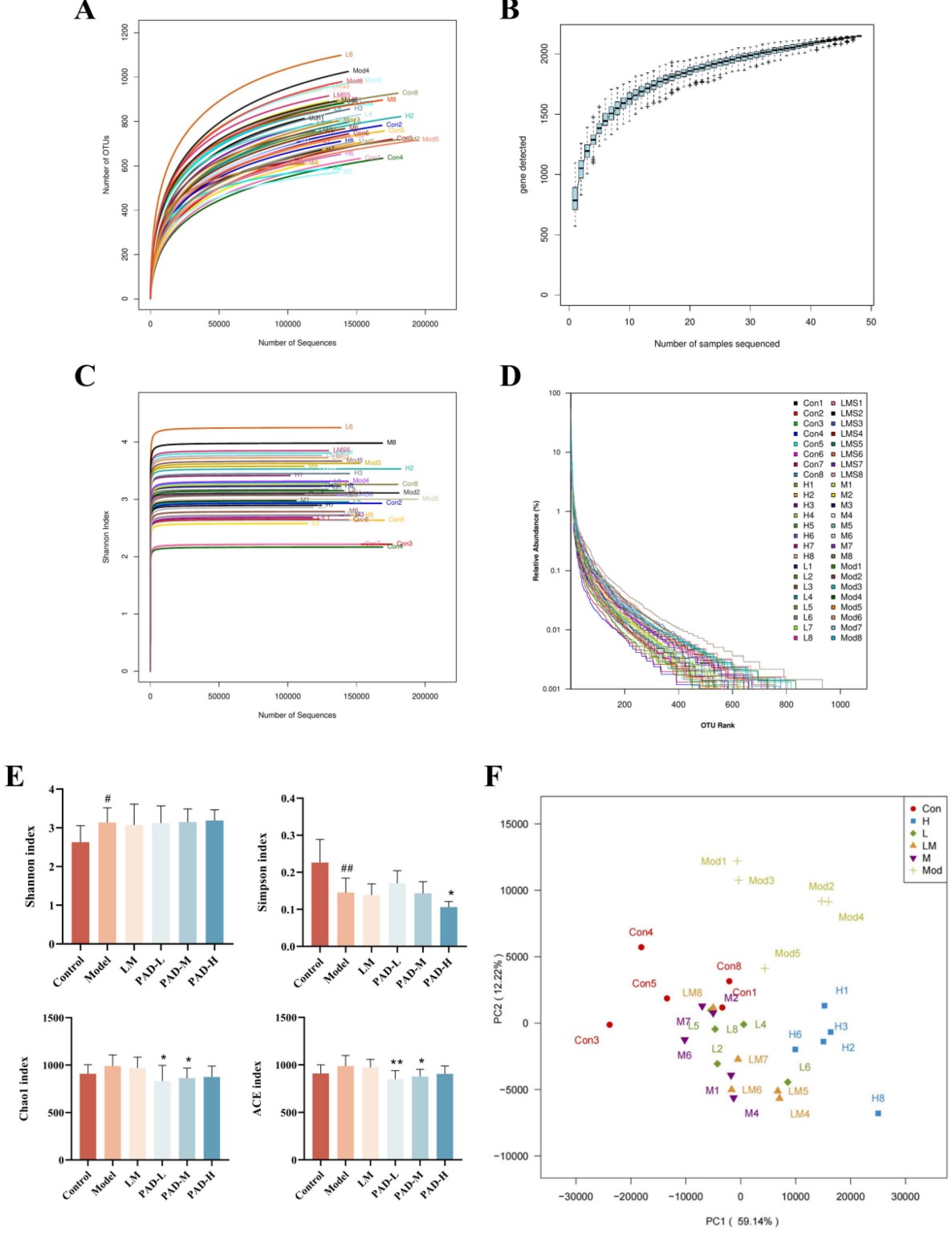

**Fig 3. PAD modulates microbial community structure and diversity. (A)** Rarefaction curves. **(B)** Species accumulation. **(C)** Shannon–Wiener curves. **(D)** Rank-abundance distribution curves. **(E)** Alpha diversity analysis of GM in rats. **(F)** PCA of the effect of PAD on GM in rats with ROUs. Data are expressed as mean ± SEM ($n = 5$). **#** $P < 0.05$ and **##** $P < 0.01$, as compared to the control group; **\*** $P < 0.05$ and **\*\*** $P < 0.01$, as compared to the model group.

between these groups (Fig 3F). However, the confounded distribution of the PAD and LM groups overlapped with that of the control, indicating that they had similar flora structures. In the beta diversity analysis, anomalous data were found in the PAD-L, control, and LM groups, which might be due to large differences in the individual samples, different sample processing times, and other factors. To ensure consistent statistical values, each group was selected such that they had the same sample size ($n = 5$).

### 3.6. PAD changes the structure and abundance of the GM at the phylum, family, and genus levels

In this study, we analyzed the structure and abundance of flora in rat fecal samples at the phylum, family, and genus levels. At the phylum level, there were 24 bacterial phyla in the GM. Bacteroidetes and Firmicutes were the main flora in the rat feces, accounting for more than 90% of the total intestinal microorganisms (Fig 4A). The ratio of Bacteroides/Firmicutes (B/F) in the model group increased, while that in the PAD groups decreased and gradually approached that in the control, in a dose-dependent manner. A total of 163 microbial families were identified at the taxonomic level. After PAD intervention in the PAD-M and PAD-H groups, the relative abundances of probiotics such as Ruminococcaceae and Bifidobacteriaceae increased, whereas those of probiotics such as Lactobacillaceae and pathogenic bacteria such as Enterococcaceae and Staphylococcaceae decreased (Fig 4B). At the level of genus taxonomy, compared to the model groups, the PAD-M and PAD-H groups displayed increased relative abundance of probiotics like *Ruminococcus* and *Bifidobacterium*, and decreased relative abundance of *Lactobacillus* and the pathogenic bacteria-like *Enterococcus* and

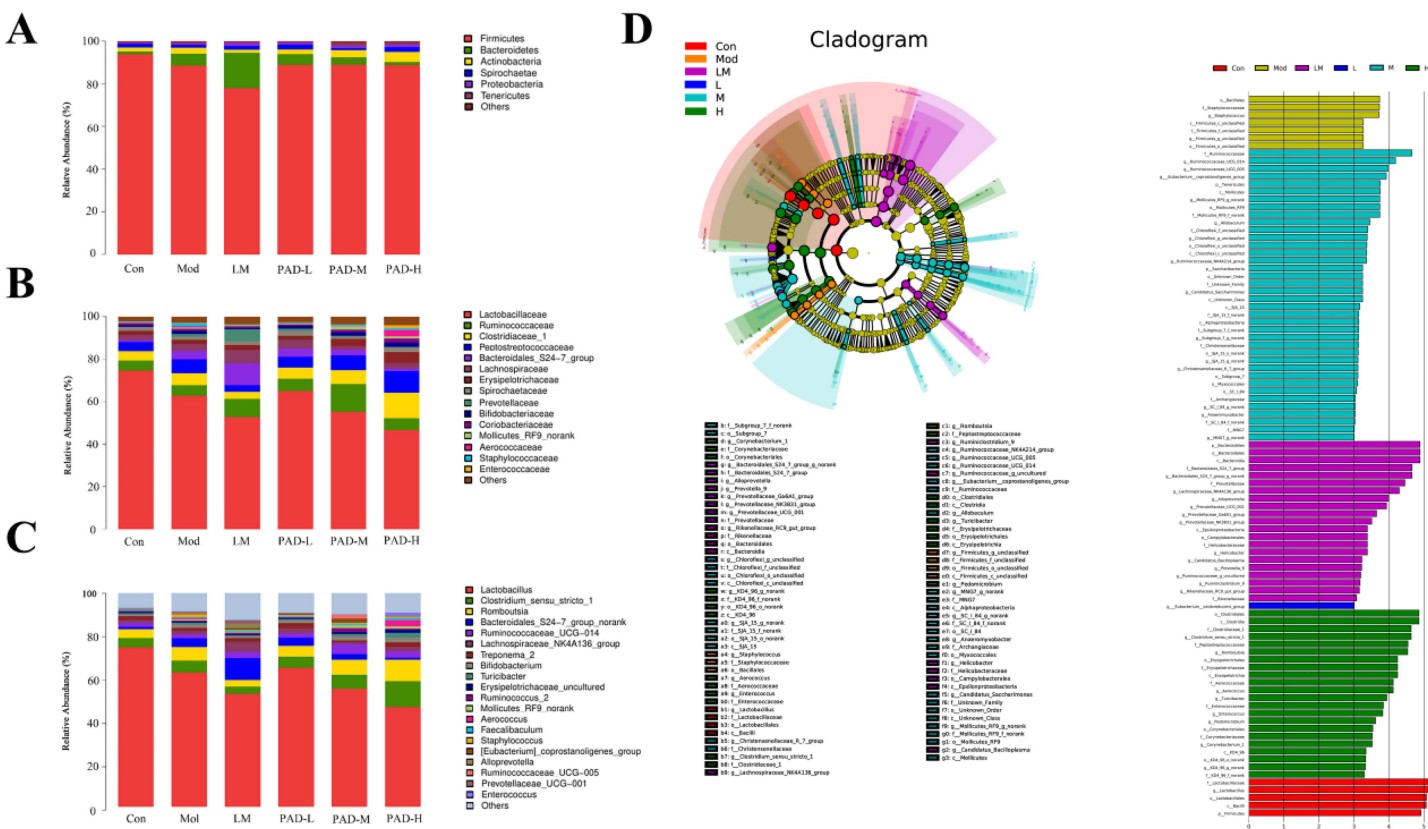

**Fig 4. Effect of PAD on the gut microbiota of rats with ROU at the phylum (A), family (B), and genus (C) levels ($n = 5$). (D)** LefSe difference analysis of PAD effects on the intestinal microflora in rats with ROU ($n = 5$).

*Staphylococcus* (Fig 4C). Thus, PAD could change the structure and abundance of the GM at the phylum, family, and genus levels.

### 3.7.  Linear discriminant analysis effect size (LefSe) difference analysis

Analysis at the taxonomic levels (phylum, family, and genus) showed that the structure and abundance of the GM differed in the different experimental groups. To further understand the biomarkers with significant differences in the structure and abundance of the GM among the rat groups, LEfSe analysis was performed to analyze the differences in microbial communities and dominant flora using evolutionary branching diagrams and Linear Discriminant Analysis histograms. As shown in Fig 4D, there were five types of GM abundances in the control group that were significantly higher than those in the other groups, and the order from large to small according to the Linear Discriminant Analysis value was as follows: Lactobacillaceae, *Lactobacillus*, Lactobacillales, Bacilli, and Firmicutes. Three types of GM abundances in the model were higher than those in the other groups, namely Bacillales, Staphylococcaceae, and *Staphylococcus*. The flora abundance was significantly higher in the PAD group than that in the other groups, and included three types: Ruminococcaceae, Clostridiales, and *Clostridia*. LEfSe analysis indicated a significant increase in the abundance of Staphylococcus in the model group, but PAD turned fecal GM composition of ROU rats similar to that observed in the control group.

### 3.8.  Correlation analysis of the inflammation degree of oral ulcers, relative abundance of the GM, and SCFA content in ROU rats

To explore the correlation between the degree of inflammation in oral ulcers, changes in the GM relative abundance, and SCFA content in rats, we performed Spearman rank correlation analysis for the expression levels of IL-6, TNF-α, IL-2, IL-10, and VEGF, relative abundance of the GM, and SCFA content. At the phylum level, the relative abundance of Proteobacteria was positively correlated with VEGF expression, that of Bacteroidetes was markedly negatively correlated with the levels of IL-2 and IL-10, and those of Actinobacteria and Tenericutes were negatively correlated with the levels of propionic acid among the SCFAs (Fig 5A, 5B). At the family level, VEGF expression was significantly positively associated with the relative abundance of Mollicutes_RF9_norank ($p < 0.05$) and significantly negatively associated with that of Lactobacillaceae ($p < 0.05$), whereas SCFA content was negatively associated with that of Staphylococcaceae (Fig 5C, 5D). At the genus level, the expression of VEGF was significantly positively correlated with the relative abundance of Mollicutes_RF9_norank and Faecalibaculum ($p < 0.05$) and negatively correlated with that of Lactobacillus ($p < 0.05$), whereas Staphylococcus had a negative association with SCFA content (Fig 5E, 5F). In addition, SCFA content was significantly positively correlated with the expression levels of IL-2 and IL-10 ($p < 0.05$) and significantly negatively associated with the expression levels of IL-6 and TNF-α ($p < 0.05$) (Fig 5G). These results indicated that PAD can improve the degree of inflammation of oral ulcers in rats by regulating the GM and its metabolites.

### 3.9.  PAD-induced histopathological changes in the colon tissue

H&E-stained sections revealed that the colon mucosa epithelium in the control group had a complete structure, without mucosal congestion and edema, and with evenly arranged villi. However, in the model group, the epithelial structure of the colon mucosa was destroyed, goblet cells were decreased, and intestinal villi were of varying lengths. The mucosa epithelial structure of the colon improved in the PAD and LM groups. In the PAD-L and LM groups, the mucosal structure tended to be complete and the intestinal villi were unevenly arranged, whereas in the PAD-M and PAD-H groups, the mucosal structure was basically complete and the intestinal villi were evenly arranged, as compared to those in the model group. There was no significant difference between the treatment and control groups in terms of goblet cells (Fig 6A).

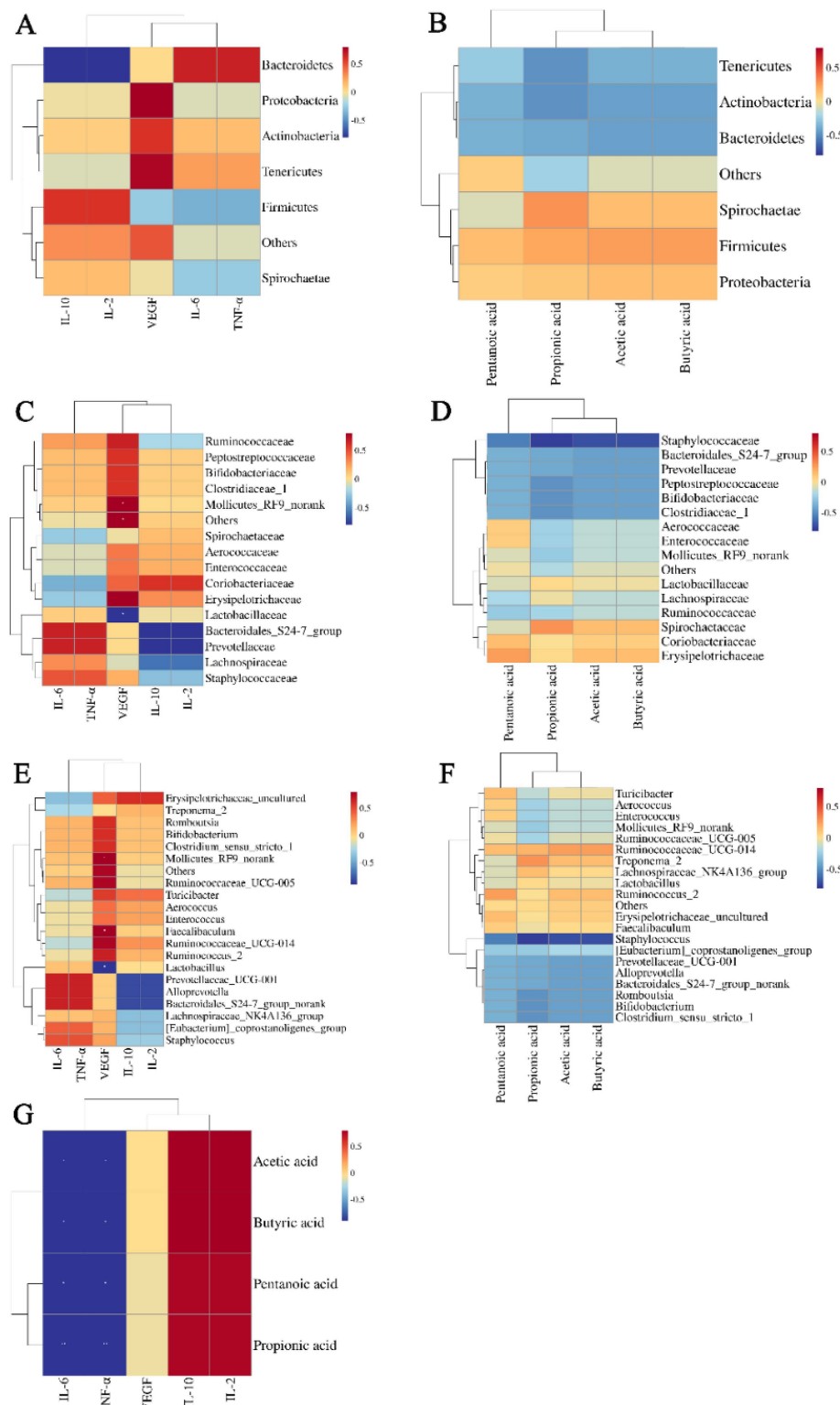

**Fig 5. Correlation analysis of inflammation degree of oral ulcer, relative abundance of gut microbiota and SCFA content in ROU rats.** Analysis of the correlation between the relative abundance of GM, expression of IL-6, TNF-α, IL-2, IL-10, and VEGF, and SCFA content at the level of phylum **(A, B)**, family **(C, D)**, and genus **(E, F)**. **(G)** Analysis of correlation between the expression of IL-6, TNF-α, IL-2, IL-10, and VEGF and SCFA content. Red, positive correlation; blue, negative correlation; darker color indicates higher degree of correlation.

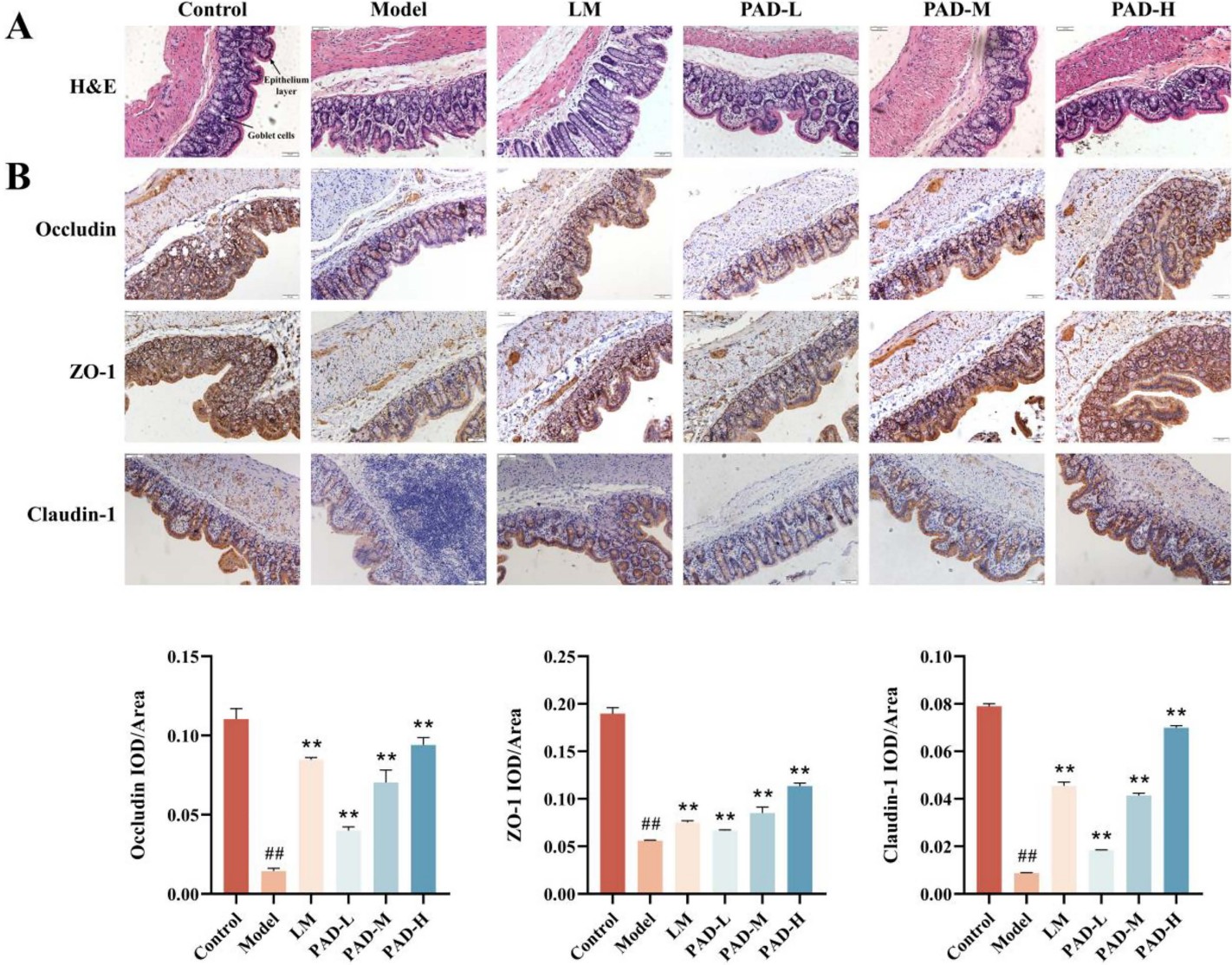

**Fig 6. Effect of PAD on histopathology and tight junction proteins in the colon of ROU rats. (A)** H&E staining analysis of the rat colons (×200), and **(B)** detection of ZO-1, occludin, and claudin-1 using immunohistochemical assay (×200) ($n = 3$). Data are expressed as mean ± SEM. # $P < 0.05$ and ## $P < 0.01$, as compared to the control group; * $P < 0.05$ and ** $P < 0.01$, as compared to the model group.

### 3.10. PAD protected the intestinal barrier by increasing the expression of tight junction proteins

The intestinal mucosa is closely associated with occludin, zonula occludens-1 (ZO-1), and claudin-1, which maintain the function and integrity of the intestinal barrier. Protein expression in the colonic mucosa was detected using immunohisto-chemistry. Occludin, ZO-1, and claudin-1 were positively expressed in the colonic mucosa of normal rats. Antigen emulsi-fier reduced occludin, ZO-1, and claudin-1 expression in the colon mucosa, but their expression in PAD-treated rats were significantly upregulated (Fig 6B).

We further verified the expression of occludin, ZO-1, and claudin-1 using western blot and RT-qPCR analyses. The pro-tein and mRNA expression levels of occludin, ZO-1, and claudin-1 decreased in the model group, but tended to be higher

in each dose group of the PAD and LM group, with the most notable changes observed in the PAD-H group (Fig 7A, 7B). These results are consistent with those of the immunohistochemistry.

ROU reduced the expression of occludin, ZO-1, and claudin-1 in the rat colon mucosa, whereas PAD reversed the effect caused by ROUs to some extent, thus showing that PAD had a protective effect on the IMB in ROU rats.

### 3.11. PAD increased the secretory immunoglobulin A (SIgA) content in the colon tissue

SIgA plays an important role in intestinal mucosa immunity and the maintenance of intestinal homeostasis. The model group had significantly decreased SIgA content in the colon tissue than the control group. Compared with the model group, all treatment groups showed an upward trend in SIgA levels in colonic tissue, but this increase was statistically significant only in the PAD-H and LM groups (Fig 8A). Thus, PAD seemed to increase the SIgA intestinal immune index in rats.

### 3.12. Correlation analysis between inflammation degree of oral ulcers and intestinal barrier

To explore the correlation between the inflammatory degree of oral ulcers and the intestinal barrier in rats, we correlated the expression levels of cytokines with those of tight junction proteins and SIgA content using Spearman rank analysis. The levels of occludin, ZO-1, claudin-1, and SIgA significantly positively correlated with the expression of IL-2 and IL-10 ($p<0.05$), but significantly negatively correlated with the expression of IL-6 and TNF-α ($p<0.05$) (Fig 8B). This suggested that the therapeutic effect of PAD on ROUs might be achieved by upregulating the protein expression of occludin, ZO-1, and claudin-1, as well as, increasing the content of the immune index SIgA, to improve the protection of the intestinal barrier in rats.

### 3.13. PAD protects the intestinal barrier through the bcl-2/bax/caspase-9 apoptosis pathway

Intestinal mucosal apoptosis can increase the intestinal mucosal permeability and decrease the intestinal barrier function. In case of intestinal mucosal damage, bcl-2 and bax proteins activate caspase-9 and induce apoptosis. To further clarify

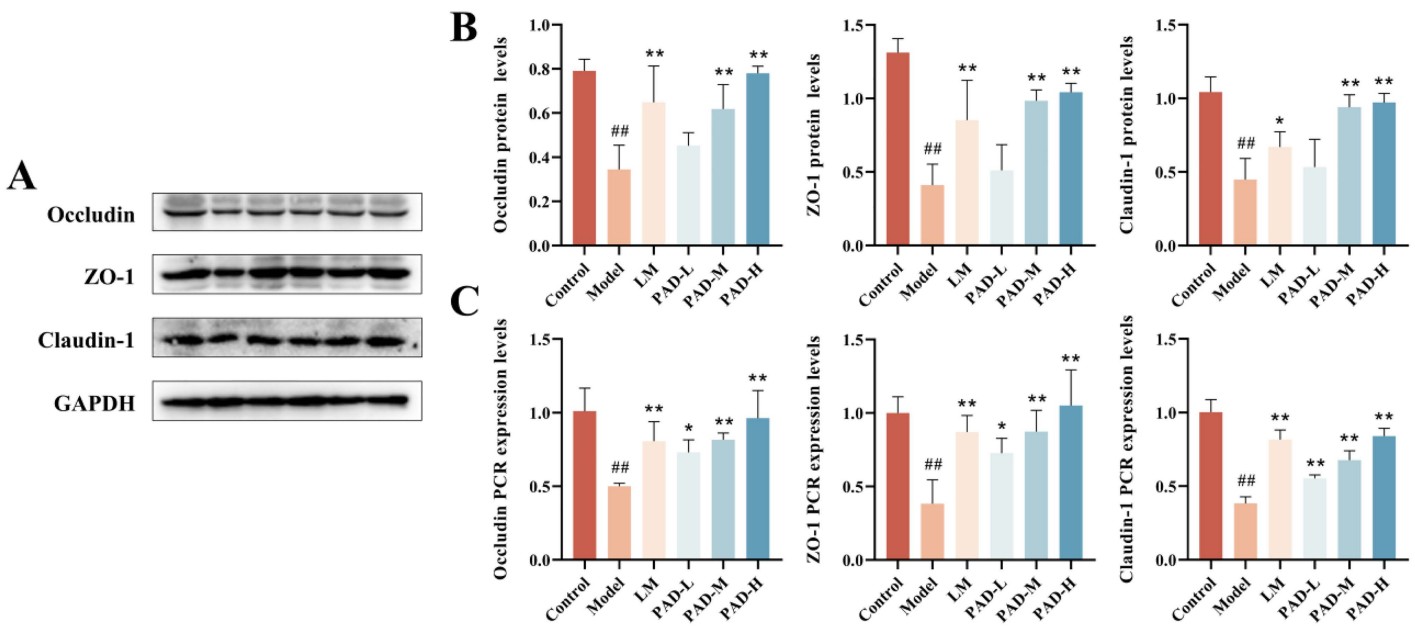

**Fig 7. Effect of the PAD on occludin, ZO-1, claudin-1 protein (A), and gene (B) expression levels in the rat colon tissue ($n=3$).** Data are expressed as mean±SEM. **#** $P<0.05$ and **##** $P<0.01$, as compared to the control group; * $P<0.05$ and ** $P<0.01$, as compared to the model group.

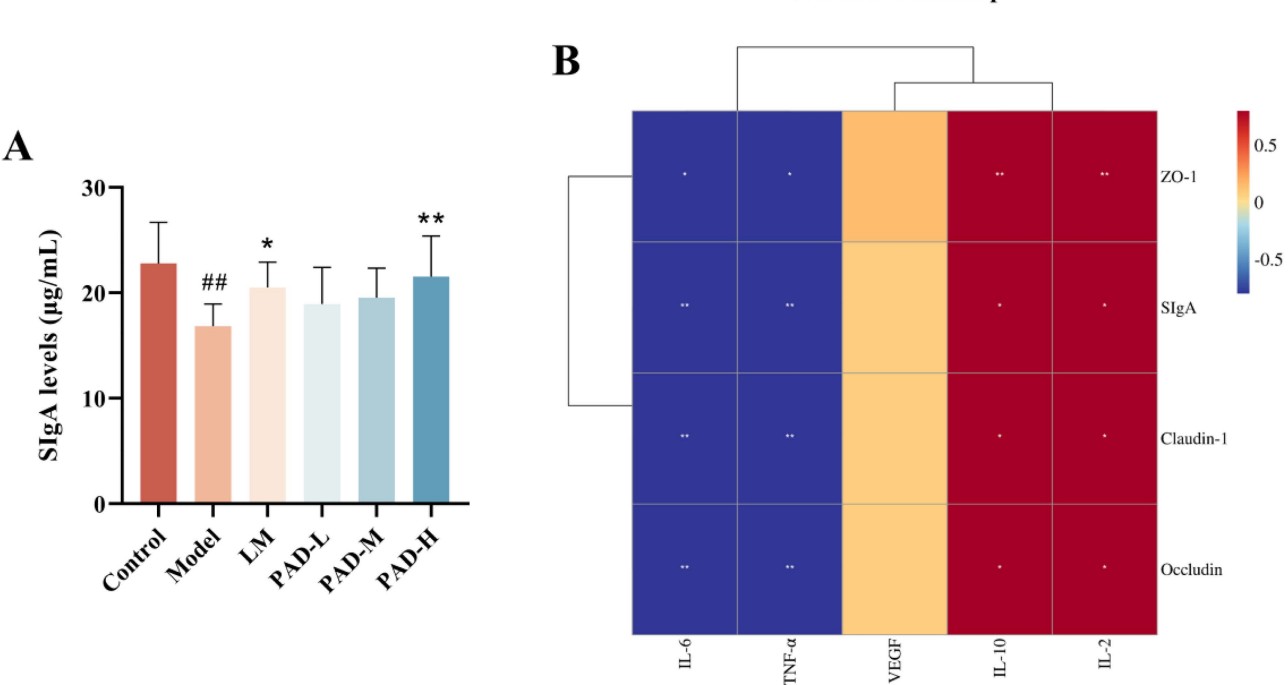

**Fig 8. SIgA level of colonic immunity and its correlation with oral ulcer inflammation and intestinal barrier integrity in rats. (A)** SIgA level of colonic immune index in rats. **(B)** Correlation analysis between the degree of oral ulcer inflammation and intestinal barrier integrity.

the effect of PAD on repair of the IMB, we measured the protein and mRNA expression levels of the apoptosis-related proteins bcl-2, bax, and caspase-9. The results of western blot and RT-qPCR showed that antigen emulsifier downregulated the protein and mRNA expression levels of bcl-2 and upregulated those of bax and caspase-9 in the intestinal barrier. In contrast, PAD increased the protein expression and mRNA levels of bcl-2 and decreased those of bax and caspase-9 in the intestines of ROU rats, and there was no significant difference in these levels between the PAD-H and control groups (Fig 9A, 9B).

## 4. Discussion

ROUs are characterized by chronic, recurrent, and segmental remission, and a clear pathological mechanism for the condition has not been identified yet [30]. Existing research has confirmed that the occurrence of ROU is closely associated with immune imbalance [28]. Its core pathological feature is T cell-mediated local mucosal immune disorder, specifically manifested as an imbalance in T lymphocyte subsets and elevated expression levels of pro-inflammatory factors such as IL-6 and TNF-α [31,32]. Most animal ROU models find it difficult to replicate the characteristics of ROUs in humans. However, the mechanism by which immunological methods induce ROU in rats may involve specific proteins acting as antigens entering the body, leading to the production of specific antibodies that trigger an immune response in the oral mucosa [33]. The primary mechanism of this immunological approach involves antigen-emulsifier-induced imbalance in rat T lymphocyte subsets and serum-related factors [34]. This imbalance subsequently leads to the development of single or multiple circular or oval localized ulcers on the rat oral mucosa, accompanied by nonspecific inflammation and gastrointestinal disorders [35]. These findings correspond to the clinical manifestations observed in human ROU patients [35]. Therefore, we mixed allogeneic oral mucosal tissue with complete Freund's adjuvant to prepare an antigen emulsifier,

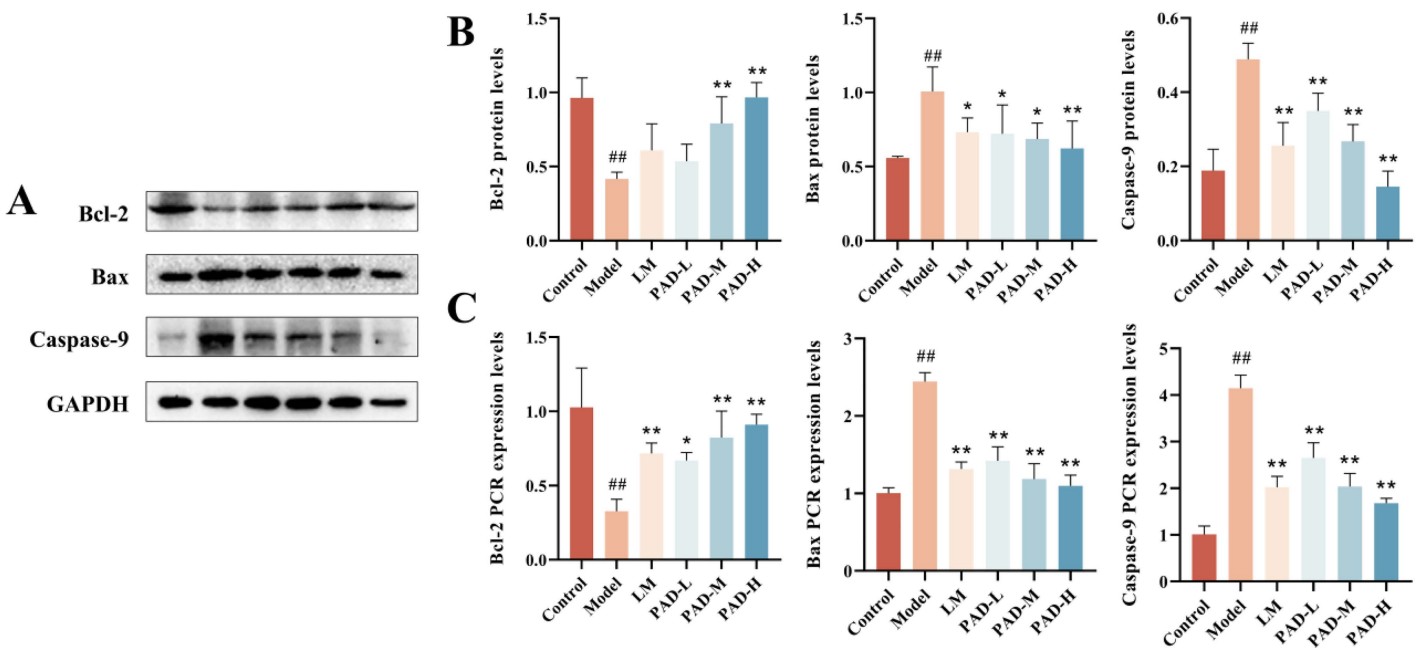

**Fig 9. Effect of PAD on bcl-2, bax, caspase-9 protein (A) and gene (B) expression levels in rat colon tissues (*n* = 3).** Data are expressed as mean ± SEM. **#** *P* < 0.05 and **##** *P* < 0.01, as compared to the control group; * *P* < 0.05 and ** *P* < 0.01, as compared to the model group.

which was then injected into animals to induce an immune response and replicate the oral ulcer model. However, immunologically induced rat ROU still suffers from limitations such as complex model establishment procedures, excessive time requirements, and inconsistent ulcer locations [35].

The ethanol extract of *P. americana* contains amino acids, amino acid metabolites, peptides, sugars, nucleosides, neurotransmitters, polyols, and organic acids [36]. The research team previously identified 40 PAD components through UPLC-Q-TOF-MS analysis, including 7 amino acids, 7 alkaloids, 6 fatty acids, 5 nucleosides, 4 peptides, 3 sugars, and 8 other compounds [37]. At present, Kangfuxin liquid is used to treat oral mucosal epithelial damage caused by tumor radiotherapy [17,38], and it is also combined with triamcinolone acetonide to treat elderly oral suboral fibrosis, reduce the level of inflammatory factors, and reduce the area of oral mucosal lesions [39]. Research methods such as network pharmacology and molecular docking have shown that peptides such as Cyclo (L-Val-L-Tyr), Cyclo (L-Pro-L-Tyr), and Cyclo (L-Tyr-L-Tyr) may be active ingredients for the treatment of oral ulcers [40]. Zhang et al. found that ethanol extract of *P. americana* could treat ulcerative colitis in rats by regulating GM, which was chemically characterized to contain nucleoside components such as inosine, uracil, and adenosine [41]. More and more natural sugars have been confirmed to have immunomodulatory activities and intestinal microbiota-regulating effects, while amino acids can be used to restore radiation-induced intestinal barrier damage in mice [42–44]. Although the composition of traditional Chinese medicine is complex, it has shown satisfactory clinical effects in complex diseases with the mode of "multi-target and multi-pathway" [45,46]. In the present study, we found that after treatment with PAD in ROU rats, SIgA, cytokines, GM and intestinal barrier were restored to varying degrees. Therefore, PAD may play a role in the treatment of ROU by jointly regulating the immunity, GM, and intestinal barrier of ROU rats through amino acids, peptides, sugars, nucleosides, and other components.

LM is an immunomodulator that regulates the human immune function, its molecular structure contains two pharmacologically active regions: an imidazole ring and a sulfur-containing region [47–49]. Currently, LM is commonly used in clinical treatment of ROUs, reducing the healing time, pain, number, and size of ROUs and preventing recurrence [50].

Studies have shown that LM can restore the immune function of T lymphocytes, B lymphocytes, monocytes, and macrophages by regulating the serum levels of IL-8 and IL-6 in patients with ROUs, thus exerting a therapeutic effect on ROUs [50–53]. Therefore, it was chosen as the positive drug for comparative test in this study. However, during the treatment of ROU with LM, patients may experience side effects such as gastrointestinal reactions and neurological reactions, with an incidence rate of 11.2%.

Inflammatory reactions are related to the severity of ROUs, and reducing inflammation is beneficial for wound healing [54]. The occurrence of ROU is often accompanied by local inflammatory responses. Studies have confirmed that the expression levels of inflammatory mediators TNF-α and IL-6 are positively correlated with the progression and severity of ROU [55,56]. In this study, TNF-α and IL-6 levels were significantly elevated in the ROU model, consistent with the pathological feature of increased proinflammatory factors in human ROU patients [57], indicating the successful establishment of the ROU model. IL-2 is a key immunomodulatory factor mainly produced by CD4$^+$ helper T cell type 1 (Th1 cells), which can promote the proliferation and activation of T lymphocytes and is an important molecule for maintaining the balance between cellular immunity and humoral immunity [58]. The results of this study indicate that serum IL-2 levels in ROU rats were significantly reduced. This decrease may be attributed to a reduction in the number of T cells in the rats' peripheral blood, which directly led to a decline in IL-2 synthesis and secretion capacity [59,60]. As a highly specific vascular endothelial cell growth factor, VEGF can accelerate vascular endothelial cell migration and proliferation, as well as granulation tissue formation, and promote oral ulcer wound repair [61]. IL-10 is an immunoregulatory cytokine that plays a crucial role throughout the immune system [62]. Therefore, PAD in this study can restore the body's immunoregulatory function by modulating cytokine levels, thereby improving rat ROU.

Millions of bacteria inhabit the human mouth, intestines, and respiratory tract, with as many as 100 trillion bacteria in the intestine [63,64]. The intestinal microenvironment is composed of GM, metabolites (SCFAs), and the intestinal mucosal immune system, which are in a dynamic balance [65]. The metabolic products of the gut microbiota, namely SCFAs (acetic acid, propionic acid, butyric acid, and valeric acid), are key mediators of immune regulation. They participate in host defense by regulating GM functions, promoting lipid metabolism, and balancing immune responses [66–70]. This study found that PAD could significantly restore the intestinal flora structure of ROU rats and increase the content of SCFAs, suggesting that it may promote oral mucosa repair by reshaping the gut-immune axis. Notably, following PAD intervention in rats with ROU, the abundance of probiotic Lactobacillus decreased. Although the total *Lactobacillus* count declined, the proliferation of *Bifidobacterium* helped maintain intestinal short-chain fatty acid levels. This suggests that functional redundancy within the gut microbiota may have offset the effects of reduced abundance in specific bacterial genera [71]. Therefore, the core objective of PAD therapy for ROU is to break the "ROU-related gut microbiota dysbiosis cycle," rather than simply modulating the abundance of single beneficial bacteria such as *Lactobacillus*. Intestinal function is affected not only by the balance of the GM but also by the integrity of the IMB [72]. The intestinal epithelium can effectively prevent the transfer of intestinal bacteria and their metabolites to parenteral tissues, mainly by relying on the mucosal barrier. According to their function, the mucosal barriers can be divided into mechanical, chemical, immunological, and biological barriers, of which mechanical barriers are the most important [73–75]. The mechanical barrier is a complete mucosa epithelial structure composed of intestinal mucosa epithelial cells and tight junctions between cells. Tight junctions are specific membrane regions composed of various transmembrane proteins such as occludin, ZO-1, and claudin-1, which can inhibit the invasion of intestinal bacteria and pathogens [76]. This study found that the expression of tight junction proteins was significantly reduced in the colon tissue of immunomodeled ROU rats, directly confirming the presence of IMB structural damage in the model rats. The intestinal immunological barrier is mainly composed of SIgA in the intestine and intestinal mucosal lymphoid tissues [77]. When the IMB is damaged, the intestinal mucosal permeability increases, SIgA secretion decreases, and lipopolysaccharide secretion increases, thus inducing an inflammatory reaction [78,79]. The findings of this study are highly consistent with the aforementioned mechanism. The significantly reduced levels of SIgA in the colonic tissue of ROU model rats further validate that a defect in the intestinal immune barrier function

indeed exists during the immune-induced ROU process. Following PAD treatment, both tight junction protein and SIgA levels recovered, suggesting that PAD may reduce intestinal permeability by enhancing mechanical barrier stability and restore SIgA-mediated mucosal immune function by regulating intestinal immune balance, thereby promoting ROU healing. To deeply explore the regulatory mechanism of PAD on IMB, we conducted an examination of the apoptotic pathway in the colonic tissues of rats. Apoptosis is a widespread process in IMB metabolism. The structural system of intestinal mucosa epithelial cells is disordered when the IMB is damaged, causing diseases that endanger human health; therefore, apoptosis plays a vital role in maintaining the stability of intestinal mucosa epithelial cells [80]. In apoptosis, the bcl-2 family mainly controls the intracellular pathway, which majorly consists of two genes, bcl-2 and bax; their mutual antagonism determines apoptosis to some extent [81]. In the bcl-2/bax signaling pathway, caspase-9 can activate caspase-3, thus causing apoptosis [82–84]. This study revealed a significant imbalance in apoptosis regulation within the colonic tissue of ROU model rats. This study revealed a significant imbalance in apoptosis regulation within the colonic tissue of ROU model rats. The abnormal expression of these apoptosis-related genes directly leads to dysfunction of colonic epithelial cells, compromising the physical defense barrier of the intestine and creating conditions conducive to pathogenic microbial invasion [85,86]. However, PAD can preferentially restore the expression of genes and proteins associated with the apoptosis pathway. The above findings suggest that PAD may promote the healing of oral ulcers in rats by repairing damage to the intestinal mucosal barrier through regulating the levels of tight junction proteins and apoptosis-related proteins, thereby modulating the gut microbiota.

This study focuses on the relationship between the oral and intestinal microenvironments, confirming that intestinal microenvironmental dysregulation impacts oral health. It also clarifies that PAD can promote ROU healing by restoring intestinal microenvironmental balance, providing insights for ROU mechanism research and the development of natural therapeutic agents. However, this study still has certain limitations, such as the failure to explore the direct association and interaction mechanisms between the oral microbiota and the GM, the small sample size, and the lack of female samples. Future research could combine animal model validation with clinical sample analysis to clarify the pathophysiological significance of GM regulation of the oral microbiota. This would provide more comprehensive experimental evidence for developing ROU prevention and treatment strategies based on "gut-oral microbiota co-regulation", thereby advancing the clinical translation of related research.

## 5. Conclusions

In summary, PAD exerts a significant improvement effect on ROU, with its mechanism involving the multi-dimensional synergy of local inflammation inhibition, intestinal microecology regulation, and intestinal barrier protection. PAD can reduce the infiltration of local inflammatory cells, downregulate the expression of pro-inflammatory factors IL-6 and TNF-α, and upregulate the levels of anti-inflammatory factors IL-2, IL-10, and pro-repair factor VEGF, thereby alleviating oral inflammation and promoting wound healing. Meanwhile, PAD is capable of regulating the diversity and abundance of GM, increasing the content of SCFAs in feces, improving colonic pathological damage, enhancing intestinal barrier function, and reducing colonic cell apoptosis. This study reveals that PAD achieves a synergistic therapeutic effect through the "gut-oral axis", providing an experimental basis and ideas for the clinical application of PAD and the development of innovative drugs for ROU.

## Supporting information

**S1 File. Graphical abstract.**
(DOCX)

**S2 File. Original western blot gel image data.**
(RAR)

## Author contributions

**Conceptualization:** Kailing Li.

**Data curation:** Kailing Li, Liping Yuan.

**Formal analysis:** Jingyu Zhang.

**Funding acquisition:** Yongshou Yang, Peiyun Xiao.

**Investigation:** Guanhua Zhao.

**Methodology:** Kailing Li, Liping Yuan.

**Project administration:** Kailing Li, Liping Yuan.

**Resources:** Yongshou Yang, Zhengchun He, Peiyun Xiao.

**Software:** Weijun Li.

**Supervision:** Kailing Li, Liping Yuan, Yongshou Yang, Zhengchun He, Peiyun Xiao.

**Validation:** Kailing Li, Liping Yuan.

**Visualization:** Zhongze Chen.

**Writing – original draft:** Kailing Li.

**Writing – review & editing:** Kailing Li.

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
