## [Decision Letter · Decision Letter 0]

10 Oct 2025

Dear Dr. Xiao,

Thank you for submitting your manuscript to PLOS ONE. After careful consideration, we feel that it has merit but does not fully meet PLOS ONE’s publication criteria as it currently stands. Therefore, we invite you to submit a revised version of the manuscript that addresses the points raised during the review process.

In addition to Reviewers' comments, please address following suggestions by the Editor:

1) Please make sure that all relevant research efforts of other groups are adequately cited in the manuscript—both in favor/support and against the current research.

2) Please ensure that the methodology is detailed and all information for reagents has been included.

3) Please ensure scale bars, labels, resolution of images are appropriate and adherent to Journal guidelines.

4) Please ensure that statistical analyses are rigorously performed for all data. Please also ensure that data shown in figures are representative of at least three independent experiments performed in replicates.

5) Please avoid/remove any overstatements in the manuscript. Please ensure that the statements are based on results obtained.

We look forward to receiving your revised manuscript.

Kind regards,

Mukul Godbole, Ph.D.

Academic Editor

PLOS ONE

Journal Requirements:

3. In your Methods, please describe any chemical analyses you performed on the PAD extract you used in this study. Alternatively, please cite another study in which the same extract has been analyzed.

“This study was supported by the Basic Research Key Projects of Science and Technology Department of Yunnan Provincial (202501AS070162) , the Joint Special Focus Program for Local Colleges and Universities in Yunnan Province (grant number 202401BA070001-007), and the Yunnan Expert Workstation (202405 AF140044).”

“This study was supported by the Basic Research Key Projects of Science and Technology Department of Yunnan Provincial (202501AS070162) , the Joint Special Focus Program for Local Colleges and Universities in Yunnan Province (grant number 202401BA070001-007), and the Yunnan Expert Workstation (202405 AF140044).”

6. We note that your Data Availability Statement is currently as follows: [All relevant data are within the manuscript and its Supporting Information files.]

7. PLOS requires an ORCID iD for the corresponding author in Editorial Manager on papers submitted after December 6th, 2016. Please ensure that you have an ORCID iD and that it is validated in Editorial Manager. To do this, go to ‘Update my Information’ (in the upper left-hand corner of the main menu), and click on the Fetch/Validate link next to the ORCID field. This will take you to the ORCID site and allow you to create a new iD or authenticate a pre-existing iD in Editorial Manager.

8. Thank you for stating the following in the Acknowledgments Section of your manuscript:

“This study was supported by the Basic Research Key Projects of Science and Technology Department of Yunnan Provincial (202501AS070162) , the Joint Special Focus Program for Local Colleges and Universities in Yunnan Province (grant number 202401BA070001-007), and the Yunnan Expert Workstation (202405 AF140044).”

“This study was supported by the Basic Research Key Projects of Science and Technology Department of Yunnan Provincial (202501AS070162) , the Joint Special Focus Program for Local Colleges and Universities in Yunnan Province (grant number 202401BA070001-007), and the Yunnan Expert Workstation (202405 AF140044).”

9. PLOS ONE now requires that authors provide the original uncropped and unadjusted images underlying all blot or gel results reported in a submission’s figures or Supporting Information files. This policy and the journal’s other requirements for blot/gel reporting and figure preparation are described in detail at https://journals.plos.org/plosone/s/figures#loc-blot-and-gel-reporting-requirements and https://journals.plos.org/plosone/s/figures#loc-preparing-figures-from-image-files. When you submit your revised manuscript, please ensure that your figures adhere fully to these guidelines and provide the original underlying images for all blot or gel data reported in your submission. See the following link for instructions on providing the original image data: https://journals.plos.org/plosone/s/figures#loc-original-images-for-blots-and-gels.

Additional Editor Comments (if provided):

Dear Authors:

Your manuscript PONE-D-25-27151 has been reviewed by multiple reviewers, who have provided some important comments and suggestions. This is an important study and provides some interesting mechanistic insights; but the manuscript requires significant changes. Therefore, we are inviting you to submit a revised version of the manuscript. Please address all the comments of all reviewers in a pointwise response letter. Please incorporate all suggestions of the reviewers in the manuscript and highlight the textual changes in blue font.

In addition, please address comments by the editor:

1) Please make sure that all relevant research efforts of other groups are adequately cited in the manuscript—both in favor/support and against the current research.

2) Please ensure that the methodology is detailed and all information for reagents has been included.

3) Please ensure scale bars, labels, resolution of images are appropriate and adherent to Journal guidelines.

4) Please ensure that statistical analyses are rigorously performed for all data. Please also ensure that data shown in figures are representative of at least three independent experiments performed in replicates.

5) Please avoid/remove any overstatements in the manuscript. Please ensure that the statements are based on results obtained.

Reviewers' comments:

Reviewer's Responses to Questions

**Comments to the Author**

1. Is the manuscript technically sound, and do the data support the conclusions?

Reviewer #1: No

Reviewer #2: Yes

Reviewer #3: Yes

Reviewer #4: Yes

Reviewer #5: Partly

Reviewer #6: Yes

2. Has the statistical analysis been performed appropriately and rigorously?

Reviewer #1: No

Reviewer #2: Yes

Reviewer #3: Yes

Reviewer #4: Yes

Reviewer #5: Yes

Reviewer #6: Yes

3. Have the authors made all data underlying the findings in their manuscript fully available?

Reviewer #1: Yes

Reviewer #2: Yes

Reviewer #3: Yes

Reviewer #4: Yes

Reviewer #5: Yes

Reviewer #6: Yes

4. Is the manuscript presented in an intelligible fashion and written in standard English?

Reviewer #1: No

Reviewer #2: Yes

Reviewer #3: Yes

Reviewer #4: Yes

Reviewer #5: Yes

Reviewer #6: Yes

Reviewer #1: 1. The author should carry out experiments to understand the chemo profile of Periplaneta americana extract. Without knowing what is contains, its not justified to consider it as therapeutic agent.

2. Dose dependent toxicity studies are required.

3. The precise molecular markers related with recurrent oral ulcers should be carried out using a model cell line as well in in vivo model study, simply apoptosis related markers is not enough.

4. There is need to select a positive control for comparison purpose.

5. The MS has been written in very poor manner.

Reviewer #2: This research is exceptionally well-designed and meticulously executed. It presents a highly engaging and valuable contribution to the therapeutic field of ulcer, employing a wide range of advanced techniques to robustly support and validate the proposed hypothesis.

Reviewer #3: The manuscript presents a well-designed study on the therapeutic effects of Periplaneta americana extract (PAD) in a rat model of recurrent oral ulcers. It provides valuable insights into PAD’s multi-target actions through the gut-oral-immune axis. The findings support PAD's potential as a natural therapeutic agent, especially in traditional Chinese medicine. The study contributes meaningfully to the field of natural compound-based therapy and mucosal immunology.

Reviewer #4: Abstract

-Some meanings of the acronyms were not expressed in this section, such as SCFAs, RT-qPCR, and WB. They should be given in their full form when they were first used in the text to increase clarity.

-I would prefer a structured abstract rather than a uniform text.

-Except for these, this section clearly summarizes the objective, methods, and results of the study.

Introduction

-While the graphical abstract well summarizes the study, the term located in the figure explanation "ameliorates" should be changed to "ameliorating" or "amelioratory" in the revised version.

-" While ROU attacks are localized and typically resolve on their own, the intense burning pain, swollen lymph nodes, general malaise, and other symptoms that come with the onset of the condition can significantly impact patients' mental well-being and quality of life"

Although I agree with the fact that the symptoms can decrease the quality of life, as a patient with ROUs, I found the term "impacting the mental health significantly" to be extravagant. For me, this disease is a mild disease without a severe, life-threatening symptoms and complications.

-"Currently, ROU is clinically treated with Western medicine, and the commonly used drugs are glucocorticoids, thalidomide, and colchicine, but they have adverse effects such as teratogenicity, dizziness, and gastrointestinal adverse reactions" In this sentence, the authors presented the ROUs similar to the clinical symptoms and treatment of Behçet's Disease. However, isolated ROUs are free from multisystemic involvement and rheumatological clinical progress. Therefore, to avoid confusion in readers, the authors should clearly define and illustrate the clinical picture of the ROUs.

-Except for these, the introduction section clearly explains the objective of the study and pathophysiological interactions between the ROUs, intestinal barrier, and Chinese traditional therapies.

Methods

-"Complete Freund’s adjuvant, butyric, propionic, butyric, and 101 pentanoic were purchased from Sigma-Aldrich (St. Louis, MO, USA)." "Butyric" is repeated in this sentence.

-"UPLC-Q-TOF-MS (Agilent, USA). Microscope (Olympus, Japan). GC-MS system 118 (Agilent, USA). QuantiFluor™-ST blue fluorescence quantitative system (Promega, 119 Madison, WI, USA). Illumina MiSeq platform (Shanghai Yuan Shen Biomedical 120 Technology Co. Ltd., Shanghai, China). Ultramicro nucleic acid and protein analyzer 121 (Thermo Fisher Scientific, Waltham, MA, USA). CFX96 Real-Time PCR Detection 122 System (Bio-Rad Laboratories, Hercules, CA, USA). Image-Pro Plus 6 software (Media 123 Cybernetics, Inc., Rockville, MD, USA). SPSS 26.0 statistical software (SPSS Inc., 124 Chicago, IL, USA)." There is no meaningful sentence in this part except brand names.

-Except for these, the methods section is very detailed and explains all of the steps performed during the study.

Results

-The authors supported their results with very detailed figures, illustrations, and demonstrations.

Discussion

-The discussion section widely explains the results of the study. Numerous references and comparisons with similar studies in the literature are available.

Conclusion

This section effectively summarizes the significance of the results and their implications for the literature and clinical practice. However, the limitations of the results should have been discussed at the end of the discussion section rather than in this section.

Generally, this is a detailed study examining the effect of PAD treatment, a component of traditional Chinese medicine, in ROUs. Several parameters related to this issue were evaluated in the study, including inflammatory cytokine levels, tissue specimens, and DNA analysis. The topic is important, and the results are novel, with the potential to impact the daily practices of clinicians. Therefore, I found this study very valuable.

Reviewer #5: Manuscript title: Periplaneta americana extract ameliorates recurrent oral ulcers in rats by enhancing the intestinal epithelial barrier and regulating gut microbiota

Manuscript ID: PONE-D-25-27151

The paper presents a comprehensive mechanism where Periplaneta americana extract (PAD) treats ROUs not just locally, but by improving gut health. It regulates the gut microbiota, increases beneficial metabolites and strengthens the intestinal barrier which in turn reduces systemic inflammation that contributes to the formation and recurrence of oral ulcers.

The manuscript shows many significant findings, while some unaddressed issues are present such as

• Discussion on the correlation between oral and gut microbiota.

• The study does not identify or isolate the bioactive compound responsible for the action which makes it difficult to standardize for therapeutic application or even just for laboratory replication.

• The results show a decrease in the Lactobacilli concentration. This is counterintuitive as Lactobacillus is generally a probiotic bacterium beneficial for gut health. The discussion does not discuss why the treatment would lead to a reduction in a beneficial genus that is abundant in the control group.

The paper provides compelling evidence for PAD’s efficacy through gut-oral axis, its primary limitation is the lack of oral microbiota data, uncharacterized crude extract and the lack of clarity whether the improved intestinal barrier is a direct result of PAD’s action or an indirect consequence of the modulated gut microbiota and increased SCFA production.

The discussion section is more like a combined version of the introduction and the results. The author has not discussed the results with other available research on ROUs or PAD as such. The discussion part could be rewritten to give a persuasive validation for the hypothesis made in the study.

The study demonstrates PAD’s effectiveness over a 7-day treatment period. A key feature of ROUs is their recurrence. While the introduction notes that Traditional Chinese medicine can inhibit recurrence rates, the short duration of the study does not allow for long term effects on preventing ROUs. Since the study is based on the hypothesis of amelioration of ROUs by enhancing gut microbiome, the sustainability of the changes in gut microbiota and intestinal barrier should be made clear. Effective arguments regarding the sustainability of the same after the treatment dosage reduction or cessation should also be mentioned to strengthen the correlations made in the study.

Queries:

1. Does the Kangfuxin liquid also work through a similar mechanism in reducing the ROUs as its raw material is also PAD and has been in use for a long time?

2. Mention the primer sequence for the internal control for the qPCR and cite a reference for the same. Also cite a reference for the RQ calculation.

3. IL-2 is seen in higher level in RAS patients and contributes to the release of TNF-α and IL-6. Why is the serum level higher in control and lower in the model group? (Section3.3, Fig. 1B)

4. Section 3.3, why IL-2 expression not determined among all the other cytokines expressed in serum? (Fig. 1C)

5. Section 3.11 & Fig 8A, SIgA significantly reduced in the model group seems not to be the right term as compared to the treatment group, the levels of SIgA are on a similar level.

6. Levamisole hydrochloride (LM), has been used as positive control in the study. Since LM is commonly used in clinical treatments of ROUs, mentioning its side effects or why PAD could be better than commercially available drugs could be included in either the introduction or the discussion section- justification required.

Major corrections:

• Periplaneta americana extract in abbreviated (PAD) form has not been mentioned prior to Line 71.

• Line 100: Butyric has been mentioned twice.

• Lines 117-124 could be written as a sentence. Mentioning each instrument separately makes it incomplete.

• Line 127: medicinal material has been mentioned twice.

Suggestion: “Briefly, 540g of Periplaneta americana (medicinal material) was crushed into a powder using a mortar and pestle.” (or whatever instrument was used to grind the insects)

• Section 2.3: The title of the section sounds vague

Suggestions: Experimental model, research model or model organism.

• Mention the diet for the model organism used in the study.

• The term model group can be misinterpreted. Standard term “negative control” can be used to indicate rats administered with saline solution.

• Fig 6A: Change “gobler cell” to “goblet cells”

• Fig legend: either mention ROUs as in Fig 1, 5, 6 or recurrent oral ulcers as mentioned in Fig 3, 4.

• Use the abbreviated form for an organism’s name throughout the manuscript after its first mention. (ex. Discussion line 477: Periplaneta americana to P. americana).

Minor corrections:

• Follow the same tense through the manuscript (ex. Line 193: sulfuric acid was added and then in the same paragraph Line 195: microporous surface is used to)

• Graphical abstract caption “ameliorates effect” to “ameliorating effects”

• Line 93: “material and methods” to materials and methods”

• Line 414: “antigen emulsifier” to “Antigen emulsifier” (Sentence case)

• Fig 1A legend: Line 635 ‘theoral mucosa’ to the oral mucosa’

• Fig 6 legend: Line 661 add spacing

Reviewer #6: Major Comments

Model justification: The antigen emulsifier-induced ROU model is not standard. The authors should provide stronger justification that this model reflects the pathophysiology of human recurrent aphthous ulcers, and discuss its limitations.

The conclusion should be tempered to emphasize association rather than causality.

Overstated conclusions: The manuscript at times suggests PAD is an “effective treatment” for ROUs. Given that the data are preclinical (rat model only), the discussion should highlight translational limitations and avoid clinical overstatements.

Clarity and redundancy: The Introduction and Discussion are long and at times repetitive. Streamlining would improve readability.

Minor Comments

• Figures and legends should be checked for clarity (some are difficult to interpret in the PDF).

• Abbreviations (PAD, ROU, SCFA, GM) should be redefined in Results for reader convenience.

• Ethical approval is mentioned, but animal welfare descriptions could be expanded (analgesia, humane endpoints).

• The limitation section briefly mentions oral flora, but further elaboration is needed (e.g., small sample size, lack of female animals).

• References: Some claims in Introduction and Discussion would benefit from more recent citations.

• Grammar and syntax require polishing for international readability.

**Do you want your identity to be public for this peer review?** For information about this choice, including consent withdrawal, please see our Privacy Policy

Reviewer #1: No

Reviewer #2: **Yes:** Shweta Sharma

Reviewer #3: No

Reviewer #4: **Yes:** Assoc. Prof. Kubilay İşsever

Reviewer #5: **Yes:** S Janarthanan

Reviewer #6: No

---

## [Author Response · Author response to Decision Letter 1]

21 Nov 2025

Response to Reviewers

Manuscript Number: PONE-D-25-27151

Periplaneta americana extract ameliorates recurrent oral ulcers in rats by enhancing the intestinal epithelial barrier and regulating gut microbiota

Dear Reviewers,

Thank you very much for your valuable, detailed, and professional comments and queries about the scientific content in this revised manuscript, which gave us clear direction and scientific guidance during the revision process. All the authors and I have made detailed, targeted, and comprehensive revisions and answers to the questions raised by the reviewers together in the relevant manuscript. The revised manuscript was marked in blue font. The responses to the suggestions made by the editors and reviewers are as follows:

Editor:

1. About the question “Please make sure that all relevant research efforts of other groups are adequately cited in the manuscript—both in favor/support and against the current research.”

Reply: Thank you for your valuable suggestions—this requirement is crucial for enhancing the academic rigor and completeness of the research. Firstly, we have systematically searched recent research literature in the relevant field, comprehensively sorted out academic achievements that support or differ from the conclusions of this study, and ensured that no important related research is omitted. Secondly, we will supplement and standardize citations in relevant sections such as the introduction and discussion: we will respond to studies supporting this research, objectively present studies with different viewpoints, and analyze the possible reasons for the discrepancies in combination with the research design and results of this study. Finally, all supplementary citations will strictly follow the reference format requirements of the target journal to ensure that the citations are standardized and accurate. We will earnestly implement the above work, striving to make the manuscript more comprehensive in academic perspective and more sufficient in argumentation. Thank you again for your rigorous review and careful guidance.

2. About the question “Please ensure that the methodology is detailed and all information for reagents has been included.”

Reply: Thank you for your rigorous suggestions. Detailed and comprehensive research methods, along with complete reagent information, are the core foundation for ensuring the reproducibility and scientificity of the research. We have conducted a rigorous review of every item to ensure no omissions or errors, thereby comprehensively enhancing the standardization and transparency of the research methods. Thank you for your careful review, which will provide an important guarantee for the credibility of the research results.

3. About the question “Please ensure scale bars, labels, resolution of images are appropriate and adherent to Journal guidelines.”

Reply: Thank you for your careful reminder. We have conducted a thorough check of the scale bar settings for all experimental images, ensuring that the scale bar calibrations are consistent with the image magnification and clearly labeled at an appropriate position on each image. We have standardized the image labeling information, unified the font, font size, and labeling format, and ensured that key information such as group designations and indicators is clear and unambiguous. In accordance with the journal’s image resolution requirements, we have inspected and optimized all images. If any images fail to meet the resolution standard, they will be re-exported or reprocessed to ensure compliance with the journal’s publication criteria.

4. About the question “Please ensure that statistical analyses are rigorously performed for all data. Please also ensure that data shown in figures are representative of at least three independent experiments performed in replicates.”

Reply: Thank you for your key suggestions regarding the rigor of statistical data analysis and experimental reproducibility, which are of paramount importance for ensuring the reliability and scientific validity of the research results. During the comprehensive review of the entire manuscript, we have systematically implemented the relevant requirements and conducted rigorous verification, effectively ensuring the transparency of the research. Thank you again for your rigorous review, which has further enhanced the scientific rigor and standardization of the manuscript.

5. About the question “Please avoid/remove any overstatements in the manuscript. Please ensure that the statements are based on results obtained.”

Reply: Thank you for your crucial reminder. We have conducted multiple verifications to ensure that all statements in the manuscript are fully based on the experimental results, without any exaggerated or false content. Thank you again for your professional review, which will help us more accurately present the research value and uphold the objectivity and rigor of academic research.

Reviewer #1:

1. About the question “The author should carry out experiments to understand the chemo profile of Periplaneta americana extract. Without knowing what is contains, its not justified to consider it as therapeutic agent..”

Reply: Thank you very much for your interest in PAD bioactive components and standardization issues. Your feedback provides valuable guidance for us to more clearly convey the research background and value. We have supplemented the discussion section with 40 PAD (Periplaneta americana extract) components previously identified by the research team through UPLC-Q-TOF-MS analysis. These include 7 amino acids, 7 alkaloids, 6 fatty acids, 5 nucleosides, 4 peptides, 3 sugars, and 8 other compounds (Page 23, lines 510-512). Additionally, formulations primarily composed of PAD (Kangfuxin Liquid) have been applied in the treatment of oral mucosal epithelial damage caused by radiation therapy for tumors. The core objective of this study is to address the current situation where its clinical efficacy is well-established, but its mechanism of action remains unclear. We aim to delve into the specific pathways through which it improves oral ulcers by regulating the GM, thereby filling the gap in mechanism research. Thank you once again for your meticulous review and professional feedback, which has enabled the manuscript to more fully convey its value to PAD-related research.

2. About the question “Dose dependent toxicity studies are required.”

Reply: Thank you very much for pointing out the omissions in our manuscript. At present, preparations based on PAD (including Kangfuxin Liquid and toothpaste) have been widely used in clinical treatment and daily care scenarios. In addition, we previously conducted acute toxicity tests on this extract, and the results showed that the maximum tolerated dose (MTD) of PAD in mice exceeds 40 g/kg, with no toxicity observed and a good safety profile. Relevant experimental results are shown in the figure below:

3. About the question “The precise molecular markers related with recurrent oral ulcers should be carried out using a model cell line as well in in vivo model study, simply apoptosis related markers is not enough.”

Reply: We sincerely appreciate your insightful and constructive key recommendations. We fully acknowledge the limitations of relying solely on apoptosis-related markers in existing research, and your perspective provides a clear direction for our subsequent studies. Moving forward, we may design plans to further deepen the investigation of relevant molecular mechanisms through cellular experiments, thereby addressing the shortcomings of current research.

4. About the question “There is need to select a positive control for comparison purpose.”

Reply: Thank you very much for your comment. The positive control drug in this study is levamisole hydrochloride (LM) tablets (Page 8, lines 158 and 174), and we have explained the rationale for selecting this drug as the positive control in the Discussion section (Page 24, lines 531-541).

5. About the question “The MS has been written in very poor manner.”

Reply: We sincerely appreciate your frank feedback and rigorous review. We have comprehensively sorted out the logical structure, academic expressions, and detailed norms of the manuscript, focusing on optimizing the coherence, accuracy, and fluency of the content. We have rectified possible omissions and inappropriate points one by one, striving to significantly improve the manuscript quality and live up to your review efforts. Thank you again for your critical comments and corrections, which are of vital importance for us to improve the research results and enhance the manuscript quality.

Reviewer #2:

About the question “This research is exceptionally well-designed and meticulously executed. It presents a highly engaging and valuable contribution to the therapeutic field of ulcer, employing a wide range of advanced techniques to robustly support and validate the proposed hypothesis.”

Reply: We sincerely appreciate your high recognition and valuable feedback on this research. Your affirmation not only validates the value of this research within the field, but also strengthens our determination to continue delving deeply into this area and promoting the transformation of research results into practical applications. It also injects confidence into our future exploration of cross-technology integration applications. This undoubtedly will become an important driving force for us to further improve the paper and deepen related research in the future.

Reviewer #3:

About the question “The manuscript presents a well-designed study on the therapeutic effects of Periplaneta americana extract (PAD) in a rat model of recurrent oral ulcers. It provides valuable insights into PAD’s multi-target actions through the gut-oral-immune axis. The findings support PAD's potential as a natural therapeutic agent, especially in traditional Chinese medicine. The study contributes meaningfully to the field of natural compound-based therapy and mucosal immunology.”

Reply: We sincerely thank you for taking the time to carefully and deeply review this research and for your high praise and precise evaluation. Your feedback not only shows our deep understanding of the research content, but also greatly encourages our team to receive professional recognition for our previous efforts. We will carry forward your recognition and continue to delve deeply into the fields of natural drug therapy and mucosal immunology, striving to produce more valuable research results.

Reviewer #4:

Abstract

1. About the question “Some meanings of the acronyms were not expressed in this section, such as SCFAs, RT-qPCR, and WB. They should be given in their full form when they were first used in the text to increase clarity.”

Reply: Thank you very much for your valuable suggestions. We have carefully checked the usage of all abbreviations in the manuscript and supplemented the full names of abbreviations such as SCFAs, RT-qPCR, and WB (Page 2, lines 25-34). We ensured that all abbreviations followed the rule of “the complete form is indicated for the first occurrence”, providing clearer guidance for readers to understand the research content.

2. About the question “I would prefer a structured abstract rather than a uniform text.”

Reply: We sincerely appreciate your valuable suggestions, which have provided crucial guidance for enhancing the manuscript's quality. We sincerely appreciate your valuable suggestions, which have provided crucial guidance for enhancing the manuscript's quality. Based on your recommendations, we have revised the original narrative abstract into a structured abstract. By clearly delineating the research background and objectives, methods, results, and conclusions, we have organized key information more systematically to enable readers to quickly grasp the core content of the study (Page 2, lines 15, 21, 24, 31, and 38).

3. About the question “Except for these, this section clearly summarizes the objective, methods, and results of the study.”

Reply: Thank you very much for your recognition of this section. During the writing process, we have always strived to present the core information of the research in a more intuitive way. At the same time, we have also taken note of the other areas for improvement you mentioned earlier and have made the necessary revisions. While maintaining the clear and concise presentation of this section, we have further refined the details to ensure that the overall content is more rigorous and comprehensive.

Introduction

1. About the question “While the graphical abstract well summarizes the study, the term located in the figure explanation “ameliorates” should be changed to “ameliorating” or “amelioratory” in the revised version.”

Reply: Thank you very much for your recognition of the Graphical Abstract and for your valuable suggestions regarding the revision of terminology. We have changed “ameliorates” in the legend description to “amelioratory” to ensure that the term usage and grammatical logic are more consistent (Page 3, line 48), while maintaining the clarity of the legend description in Graphical Abstract.

2. About the question “ ‘While ROU attacks are localized and typically resolve on their own, the intense burning pain, swollen lymph nodes, general malaise, and other symptoms that come with the onset of the condition can significantly impact patients' mental well-being and quality of life.’ Although I agree with the fact that the symptoms can decrease the quality of life, as a patient with ROUs, I found the term "impacting the mental health significantly" to be extravagant. For me, this disease is a mild disease without a severe, life-threatening symptoms and complications.”

Reply: Thank you very much for your valuable suggestions. Your opinions will enable us to improve the accuracy of our expressions and better align with the actual feelings of the patients. We have revised the original phrasing, replacing “While ROU attacks are localized and typically resolve on their own, the intense burning pain, swollen lymph nodes, general malaise, and other symptoms that come with the onset of the condition can significantly impact patients' mental well-being and quality of life” with “While ROU attacks are localized and typically resolve on their own, the intense burning pain, swollen lymph nodes, general malaise, and other symptoms that accompany these attacks still interfere to some extent with the patient's daily life and have an impact on their quality of life” (Page 4, lines 56-60). Thank you once again for your professional feedback from a patient's perspective. This helps our content better align with clinical practice and the authentic patient experience.

3. About the question “ ‘Currently, ROU is clinically treated with Western medicine, and the commonly used drugs are glucocorticoids, thalidomide, and colchicine, but they have adverse effects such as teratogenicity, dizziness, and gastrointestinal adverse reactions’. In this sentence, the authors presented the ROUs similar to the clinical symptoms and treatment of Behçet's Disease. However, isolated ROUs are free from multisystemic involvement and rheumatological clinical progress. Therefore, to avoid confusion in readers, the authors should clearly define and illustrate the clinical picture of the ROUs.”

Reply: Thank you very much for your valuable suggestions. We have clearly outlined the clinical symptoms of ROU in the introduction section to ensure that the relevant background information is comprehensive and well-defined (Page 4, lines 56-60).

4. About the question “Except for these, the introduction section clearly explains the objective of the study and pathophysiological interactions between the ROUs, intestinal barrier, and Chinese traditional therapies.”

Reply: We sincerely appreciate your recognition of the Introduction section of our manuscript. This recognition not only serves as encouragement for our writing efforts, but also provides us with crucial confidence to further refine the manuscript. We always maintain a rigorous attitude to polish the manuscript.

Method

1. About the question “ ‘Complete Freund’s adjuvant, butyric, propionic, butyric, and 101 pentanoic were p

---

## [Decision Letter · Decision Letter 1]

22 Dec 2025

Periplaneta americana extract ameliorates recurrent oral ulcers in rats by enhancing the intestinal epithelial barrier and regulating gut microbiota

PONE-D-25-27151R1

Dear Dr. Xiao,

We’re pleased to inform you that your manuscript has been judged scientifically suitable for publication and will be formally accepted for publication once it meets all outstanding technical requirements.

Kind regards,

Mukul Godbole, Ph.D.

Academic Editor

PLOS One

Additional Editor Comments (optional):

Based on reviewer comments and suggestion, the manuscript can be accepted for publication.

Reviewers' comments:

Reviewer's Responses to Questions

**Comments to the Author**

Reviewer #3: All comments have been addressed

Reviewer #4: All comments have been addressed

Reviewer #6: (No Response)

2. Is the manuscript technically sound, and do the data support the conclusions?

Reviewer #3: Yes

Reviewer #4: Yes

Reviewer #6: Yes

3. Has the statistical analysis been performed appropriately and rigorously?

Reviewer #3: Yes

Reviewer #4: Yes

Reviewer #6: Yes

4. Have the authors made all data underlying the findings in their manuscript fully available?

Reviewer #3: Yes

Reviewer #4: Yes

Reviewer #6: Yes

5. Is the manuscript presented in an intelligible fashion and written in standard English?

Reviewer #3: Yes

Reviewer #4: Yes

Reviewer #6: Yes

Reviewer #3: (No Response)

Reviewer #4: I sincerely thank all the authors for their detailed explanations and corrections to the reviewer recomendations. All my recommendations are adressed and corrected in the revised version of the manuscipt.

Reviewer #6: (No Response)

**Do you want your identity to be public for this peer review?** For information about this choice, including consent withdrawal, please see our Privacy Policy

Reviewer #3: No

Reviewer #4: **Yes:** Associated Professor Kubilay İşsever, MD

Reviewer #6: No

---

## [Editor Report · Acceptance letter]

PONE-D-25-27151R1

PLOS One

Dear Dr. Xiao,

I'm pleased to inform you that your manuscript has been deemed suitable for publication in PLOS One. Congratulations! Your manuscript is now being handed over to our production team.

Kind regards,

on behalf of

Dr. Mukul Godbole

Academic Editor

PLOS One